# Punicalagin’s Protective Effects on Parkinson’s Progression in Socially Isolated and Socialized Rats: Insights into Multifaceted Pathway

**DOI:** 10.3390/pharmaceutics15102420

**Published:** 2023-10-04

**Authors:** Hoda A. Salem, Karema Abu-Elfotuh, Sharifa Alzahrani, Nermin I. Rizk, Howaida S. Ali, Nehal Elsherbiny, Alhanouf Aljohani, Ahmed M. E. Hamdan, Panneerselvam Chellasamy, Nada S. Abdou, Ayah M. H. Gowifel, Alshaymaa Darwish, Osama Mohamed Ibrahim, Zakaria Y. Abd Elmageed

**Affiliations:** 1Department of Pharmacy Practice, Faculty of Pharmacy, University of Tabuk, Tabuk 71491, Saudi Arabia; a_hamdan@ut.edu.sa; 2Department of Clinical Pharmacy, Faculty of Pharmacy, Al-Azhar University, Cairo 11884, Egypt; karimasoliman.pharmg@azhar.edu.eg; 3Department of Pharmacology, Faculty of Medicine, University of Tabuk, Tabuk 71491, Saudi Arabia; hsalama@ut.edu.sa (H.S.A.); shalzahrani@ut.edu.sa (S.A.); 4Medical Physiology Department, Faculty of Medicine, Menoufia University, Menouf 32952, Egypt; nermeen.rizk.12@med.menofia.edu.eg; 5Department of Pharmacology, Faculty of Medicine, Assiut University, Assiut 71515, Egypt; 6Department of Pharmaceutical Chemistry, Faculty of Pharmacy, University of Tabuk, Tabuk 71491, Saudi Arabia; nelsherbiny@ut.edu.sa; 7Department of Biochemistry, Faculty of Pharmacy, Mansoura University, Mansoura 35516, Egypt; 8Faculty of Pharmacy, University of Tabuk, Tabuk 71491, Saudi Arabia; 381011107@stu.ut.edu.sa; 9Department of Biology, Faculty of Science, Tabuk University, Tabuk 71491, Saudi Arabia; ppallar@ut.edu.sa; 10Faculty of Medicine, Misr University for Science and Technology (MUST), Giza 11556, Egypt; nada.abdou@etu.u-bordeaux.fr; 11Pharmacology and Toxicology Department, Faculty of Pharmacy, Modern University for Technology and Information (MTI), Cairo 11571, Egypt; ayah.gowifel@pharm.mti.edu.eg; 12Biochemistry Department, Faculty of Pharmacy, Sohag University, Sohag 82524, Egypt; alshaymaa.darwish@pharm.sohag.edu.eg; 13Clinical Pharmacy Department, Faculty of Pharmacy, University of Tanta, Tanta 31527, Egypt; osama.mohamed.ibrahim@pharm.tanta.edu.eg; 14Department of Pharmacology, Edward Via College of Osteopathic Medicine, University of Louisiana at Monroe, Monroe, LA 71203, USA; zelmageed@ulm.vcom.edu

**Keywords:** Parkinson’s disease, social isolation, PUN, HMGB1/RAGE/TLR4/NF-ᴋB/NLRP3, PI3K/AKT/GSK-3β/CREB, AMPK/SIRT-1

## Abstract

Parkinson’s disease (PD) is a gradual deterioration of dopaminergic neurons, leading to motor impairments. Social isolation (SI), a recognized stressor, has recently gained attention as a potential influencing factor in the progress of neurodegenerative illnesses. We aimed to investigate the intricate relationship between SI and PD progression, both independently and in the presence of manganese chloride (MnCl_2_), while evaluating the punicalagin (PUN) therapeutic effects, a natural compound established for its cytoprotective, anti-inflammatory, and anti-apoptotic activities. In this five-week experiment, seven groups of male albino rats were organized: G1 (normal control), G2 (SI), G3 (MnCl_2_), G4 (SI + MnCl_2_), G5 (SI + PUN), G6 (MnCl_2_ + PUN), and G7 (SI + PUN + MnCl_2_). The results revealed significant changes in behavior, biochemistry, and histopathology in rats exposed to SI and/or MnCl_2_, with the most pronounced effects detected in the SI rats concurrently exposed to MnCl_2_. These effects were associated with augmented oxidative stress biomarkers and reduced antioxidant activity of the Nrf2/HO-1 pathway. Additionally, inflammatory pathways (HMGB1/RAGE/TLR4/NF-ᴋB/NLRP3/Caspase-1 and JAK-2/STAT-3) were upregulated, while dysregulation of signaling pathways (PI3K/AKT/GSK-3β/CREB), sustained endoplasmic reticulum stress by activation PERK/CHOP/Bcl-2, and impaired autophagy (AMPK/SIRT-1/Beclin-1 axis) were observed. Apoptosis induction and a decrease in monoamine levels were also noted. Remarkably, treatment with PUN effectively alleviated behaviour, histopathological changes, and biochemical alterations induced by SI and/or MnCl_2_. These findings emphasize the role of SI in PD progress and propose PUN as a potential therapeutic intervention to mitigate PD. PUN’s mechanisms of action involve modulation of pathways such as HMGB1/RAGE/TLR4/NF-ᴋB/NLRP3/Caspase-1, JAK-2/STAT-3, PI3K/AKT/GSK-3β/CREB, AMPK/SIRT-1, Nrf2/HO-1, and PERK/CHOP/Bcl-2.

## 1. Introduction

Parkinson’s disease (PD) is a gradual neurodegenerative illness manifests as muscle rigidity, tremors, and bradykinesia [1]. These features are related to Lewy Bodies and the degeneration of dopaminergic neurons in the substantia nigra pars compacta (SNpc). The key component of the Lewy Bodies is α-synuclein (α-Syn) a neuronal protein whose misfolding can cause neurotoxicity and apoptosis of neuronal cells [2]. Although the pathophysiology of PD is still poorly understood, multiple variables, including oxidative stress, neuroinflammation, mitochondrial dysfunction, endoplasmic reticulum (ER) stress, neuronal apoptosis, and defective autophagy, have been linked to PD development [3,4,5].

Oxidative stress and neuroinflammation are interconnected processes that can mutually stimulate or repress each other [6]. Oxidative stress serves as a starting point in the pathogenesis of neurodegenerative illnesses. Notably, the high mobility group box 1 protein (HMGB1) has a crucial role in initiating neuroinflammation. During neurodegenerative conditions, HMGB1 is overexpressed and further reacts with specific receptors such as receptor for advanced glycation end products (RAGE) and toll-like receptor 4 (TLR4). This interaction induces the nuclear factor-kappa B (NF-κB) pathway. Subsequently, NF-κB activation clues to the initiation of the NOD-, LRR- and pyrin domain-containing protein 3 (NLRP3) inflammasome and pro-inflammatory mediators, including tumour necrosis factor-alpha (TNF-α), interleukin-1beta (IL-1β), and IL-6. These inflammatory molecules are connected to the development of neuroinflammation and ultimately neuronal apoptosis [7]. Upon NLRP3 activation, it initiates the activation of caspase-1 through its triggering the apoptosis-associated speck-like protein containing a caspase-recruitment domain (ASC). This process facilitates the maturation of IL-1β, thereby intensifying the inflammatory response [8]. Janus kinase-2 (JAK-2)/signal transducer and transcription-3 (STAT-3) pathway is activated because of pro-inflammatory mediators, particularly IL-6. JAK-2 mediates the STAT-3 phosphorylation, causing an astroglial activation and exacerbating inflammatory response in neurodegenerative diseases [9].

Nuclear factor-erythroid 2-related factor 2 (Nrf2) emerges as a central transcription factor associated with oxidative stress and neuroinflammation. Nrf2 effectively modulates the expression of genes encoding antioxidant enzymes and proteins, including heme-oxygenase-1 (HO-1). Nrf2/HO-1 pathway acts as a neuroprotective in various neurological conditions. Activation of this pathway has a significant impact on counteracting oxidative stress, impeding neurodegeneration, and attenuating neuroinflammation through the repression of NF-κB. Consequently, the regulation of Nrf2/ HO-1 holds a therapeutic potential for combating the deleterious consequences of oxidative stress and neuroinflammation in diverse neurological disorders [6].

Moreover, the embarrassment of the phosphoinositide 3-kinase (PI3K)/protein kinase B (AKT) pathway has been related to the aggravation of oxidative stress, neuroinflammation, and neuronal apoptosis in PD. PI3K, once activated, recruits, and activates AKT. Upon activation, AKT phosphorylates downstream targets, thereby deactivating glycogen synthase kinase 3β (GSK-3β) and triggering cyclic adenosine monophosphate (cAMP) response element-binding protein (CREB). Furthermore, GSK-3β enhances neuroinflammation by activating the NF-κB pathway and promotes oxidative stress by impairing Nrf2 signaling [10]. Additionally, CREB activation enhances the production of brain-derived neurotrophic factor (BDNF), which binds and stimulates tropomyosin receptor kinase B (TrKB), thus regulating memory, synaptic plasticity, and neuronal survival. Therefore, the AKT/GSK-3β/CREB/BDNF/TrKB pathway is a potential therapeutic target for ameliorating PD symptoms and improving patients’ outcomes [11,12].

ER is a principal organelle in proteostasis, where it controls appropriate protein folding and transport [13,14]. Misfolding of α-Syn in PD triggers an inherent cellular response recognized as ER stress. This initiates the unfolded protein response (UPR), a protective mechanism to restore proteostasis or induce poptosis if restoration fails [15,16]. When the ER encounters stress, the glucose-regulated protein 78 (GRP78) dissociates from the protein kinase RNA-like ER kinase (PERK) and binds to the misfolded proteins, thereby activating PERK and facilitating subsequent cellular events [17]. ER stress and subsequent initiation of UPR have been implicated in the degeneration of dopamine-producing neurons [18]. During α-synucleinopathy, prolonged ER stress occurs when the UPR cascade fails to restore ER homeostasis and ER stress cannot be alleviated; therefore, over activation of UPR diverts toward the apoptotic pathway, as PERK upregulates the expression of pro-apoptotic C/EBP homologous protein (CHOP) to mediate the ER stress-stimulated apoptotic pathways [6,19]. Excessive and/or sustained ER stress may possess a vital role in PD pathogenesis [16]. Therefore, hampering ER stress could mitigate the dopaminergic neuronal apoptosis and enhance neuronal survival in PD.

The balance between autophagy and apoptosis is disrupted during PD, with enhanced apoptosis and impaired autophagy, leading to the accumulation of abnormal misfolded α-syn protein, with progressive loss of dopaminergic neurons [20]. Undeniably, autophagy is an indispensable process that selectively gets rid of misfolded proteins and injured organelles, hence averting neuronal injury and stimulating neuronal survival [21]. The AMP-enhanced protein kinase (AMPK)/Sirtuin-1 (SIRT-1) pathway is the major pathway responsible for regulating neural survival, autophagy, apoptosis, and cellular metabolism [22]. By leveraging its influence through SIRT-1, AMPK effectively activates autophagy, thereby mitigating PD by inhibiting misfolded α-syn protein accumulation. Additionally, SIRT-1 plays a pivotal role by stimulating the expression of diverse downstream targets, thereby exerting control over cell survival and apoptosis [23]. Consequently, approaches that adjust the balance of apoptosis and autophagy have been suggested as the mainstay for amelioration of PD [24].

Manganese (Mn) is a widely recognized environmental neurotoxicant [25]. Elevated levels of Mn exposure have been linked to neurotoxic effects, including a Parkinson’s disease-like motor condition known as manganism [26]. The primary mechanisms driving Mn-induced neurotoxicity involve the initiation of oxidative stress, inflammation, and apoptosis pathways in the brain. These mechanisms disrupt neurotransmitter balance, contributing to neurobehavioral and motor changes resembling Parkinson’s disease [27]. 

It is noteworthy that social isolation (SI) serves as a prominent neurological stressor [28,29]. SI can potentially disrupt vital physiological processes, including neuronal connectivity, neuroplasticity, and stress response mechanisms [30]. Prior research has established a link between SI and reduced brain volumes, implying neuronal loss that correlates with neurodegenerative disorders [31]. SI also increases the risk of mental health problems, including depression, memory impairment and anxiety [28,29]. Also, the risk of neurodegenerative diseases can be reduced through maintaining strong social engagements and avoiding SI-mediated psychological stress. Several lines of evidence reported that depression could be a risk factor for developing neurodegenerative illnesses [32,33]. Hence, it is plausible to suggest that SI, as an independent factor, with its associated depression, can potentially modify or influence the progression of PD.

To date, available treatments only offer short-term relief from the symptoms of PD but do not effectively hinder the progression of the disease [34]. This highlights the urgent need to explore new therapeutic approaches that can halt the progression of PD. Therefore, it is necessary to investigate natural molecules obtained from plants that have antioxidant and anti-inflammatory properties to protect against neuronal damage and neurodegeneration [35]. One such compound of interest is Punicalagin (PUN), which is the major ellagitannin found in pomegranate (*Punica granatum* L.). PUN has shown antioxidant, anti-inflammatory, and anti-apoptotic actions. It has gained significant attention for its potential health benefits, including cardioprotective, antibacterial, anticancer, anti-atherosclerotic, and neuroprotective properties [36,37]. Previous studies have reported the neuroprotective actions of PUN in cerebral ischemia-reperfusion (I/R) damage [38] and Alzheimer’s disease [39], suggesting its potential as a therapeutic option for PD.

In recent times, there has been a growing focus on investigating the relationship between manganese exposure and the initiation of PD, as well as the subsequent neurotoxic outcomes. Mn is being explored as an inducer of PD in preclinical trials. This emerging field of research has provided valuable insights into the origins and advancement of Parkinson’s disease [40,41,42,43,44]. According to the aforementioned evidence, this study designed to assess the connection between SI as a psychological stressor and the progression of PD induced by MnCl_2_. The study aimed to elucidate the nature and pattern of this relationship by analysing behavioural, biochemical, and histopathological data and to explore the underlying mechanistic pathways that contribute to this interplay. Furthermore, the study examined the potential neuroprotective effects of PUN against the biochemical, behavioural, and histopathological alterations concomitant with SI and/or MnCl_2_ intoxication. Also, the investigation was designed to deliver mechanistic comprehensions into the role of various pathways, including HMGB1/RAGE/TLR4/NF-κB/NLRP3/Caspase-1, JAK-2/STAT-3, PI3K/AKT/GSK-3β/CREB, AMPK/SIRT-1, Nrf2/HO-1, and PERK/CHOP/Bcl-2, as potential contributors to these pathophysiological events.

## 2. Materials and Methods

### 2.1. Animals

Seventy adult male Sprague Dawley rats weighing between 300 and 320 g were acquired from The Nile Co. for Pharmaceuticals and Chemical Industries (Cairo, Egypt). Before beginning the experiment, the rats were provided a one-week acclimatization period. The rodents were housed in controlled conventional conditions at the Faculty of Pharmacy, Al-Azhar University’s animal house. These conditions entailed a 12-h light/dark cycle, 25 ± 1 °C temperature, and a humidity level of 50 ± 5%. The rats were given access to unlimited amounts of water and normal food pellets. All animal procedures and experimental protocols were ethically approved by the Animal Care and Use committee of the Faculty of Pharmacy, Al-Azhar University, with ethical approval number 395/2023. The handling of animals was according to the guidelines outlined in the “Guide for Care and Use of Laboratory Animals”, published by the National Institutes of Health (NIH Publications No. 8023, revised 1978).

### 2.2. Drugs and Chemicals

Manganese (Mn) chloride tetrahydrate (MnCl_2_.4H_2_O) and punicalagin were purchased from Sigma-Aldrich Chemical Co. (St. Louis, MO, USA). They were freshly dissolved in normal saline, as described [38]. All chemicals and reagents employed in this work were of the highest chemical grade.

### 2.3. Experimental Design

Seventy adult male albino rats were randomly assigned into 7 groups: 10 rats each. These groups (Gs) were as follows: G I (Control group) treated with 1 mL of normal saline orally and 0.2 mL of saline intraperitoneally (i.p.) in a group-housing condition [45]. G II (SI-group) consisted of rats kept separately in laboratory cages throughout the experimental period for 5 consecutive weeks to induce depression [46]. G III (Manganese (Mn)-group) received i.p. injections of MnCl_2_ at a dose of 10 mg/kg/day for 5 consecutive weeks to induce PD [47,48]. G IV (Mn + SI-group) comprised SI rats injected with MnCl_2_ for 5 weeks to induce depression concomitantly with PD (PDD). G V (SI + PUN group) received daily oral administration of PUN at a dose of 30 mg/kg in SI condition for 5 weeks, according to prior reports [38,49,50]. G VI (Mn + PUN-group) received intraperitoneal injections of MnCl_2_ and concurrent daily oral administration of PUN for 5 weeks. G VII (Mn + SI + PUN-group) consisted of SI rats treated with MnCl_2_ and concurrently receiving PUN for 5 weeks. The treatment was continued for five weeks [51]. All rats were subjected to behavioral tests 24 h following the last dosage at the close of the experiment. Subsequently, after an additional 24 h, the rats were anesthetized with ketamine (80 mg/kg, i.p.) and then euthanized for biochemical analyses and histopathological examinations.

### 2.4. Animal Housing Conditions

The SI rodents were isolated in laboratory cages, where each rat was individually housed, as one rat per cage for 5 weeks. The animals were kept in groups of four in identical-sized laboratory cages under group housing settings. Throughout the five weeks of the study, all rats were kept in their designated housing circumstances.

### 2.5. Behavioural Evaluations

#### 2.5.1. Open Field Test (OFT)

The rats’ locomotor activity was evaluated utilising the locomotor activity test. Each rat was carefully positioned in the middle of an open, square wooden box that was 80 × 80 × 40 cm in size and had red walls and a smooth, white polished floor. As previously stated, black lines partitioned the floor into 16 equal squares that were each 4 × 4 [52,53]. The rats were given 5 min to explore the room freely, and their behaviour was recorded by a video camera placed on top of the test box. Several behavioural parameters were observed, counted, and analyzed for statistical differences, including ambulation frequency (number of squares crossed by the rat in 3 min), grooming frequency (number of times the rat scratched its face, licked its forelimbs, fur, and genitals in 3 min), rearing frequency (number of times the rat stood stretched on its hind limbs with or without forelimb support in 3 min), and latency time (time taken by the rat to decide to move after being placed in the middle of the arena) [52,53].

#### 2.5.2. Forced Swim Test (FST)

The FST was conducted using a cylinder-shaped plexiglass apparatus measuring 40 cm tall and 20 cm in diameter. The water temperature was maintained at 23 ± 2 °C, and the water depth was set at 30 cm to prevent the rat’s paws from reaching the plexiglass floor [54]. There were a couple of stages to the experiment: training and testing. In the training stage, each rat was forced to swim for 15 min, and their activity, including swimming, climbing, and immobility, were recorded in the last 5 min. The test stage occurred 24 h later, where various escape-related behaviours, such as swimming, climbing, and immobility, were scored. Each rat was submerged in water for 6 min in this stage, and the last 5 min of their activity were recorded. A comparison with other groups was made using the total count of each behavior [55,56]. A higher score on the FST is typically interpreted as a sign of greater anxiety and depression [57].

#### 2.5.3. Y-Maze Test

The Y-maze test was carried out following the procedure described by Hritcu et al. [58]. This test evaluated the rats’ ability to explore a new environment by measuring the spontaneous alternation percentage (SAP). SAP is an indicator of spatial working memory, a form of short-term memory [59]. The apparatus employed for the test was a black wooden maze with 3 identical arms arranged in the shape of a capital Y (labeled as A, B, and C). Each arm had dimensions of 12 cm in width, 40 cm in length, and 35 cm in height. The arms were positioned at 120° angles from each other. During the test, each rodent was positioned at the center of the Y-maze. For 8 min, the researchers recorded the sequence of entries made by the rats into the three arms. The evaluation of SAP was based on the complete entry of each arm, with the rat’s hind paws fully entering the arm. The following formula was used to compute the spontaneous alternation percentage (SAP): SAP = [number of alternations/(total arm entries − 2)] × 100 [60].

### 2.6. Tissue Sample Collection and Preparation

After euthanizing the rats, their brains were carefully extracted and then rinsed with saline. The rodents within each group were subsequently divided into 2 sets of rats. In the first set (n = 4). For histological investigation, the brains in the first set (n = 4) were preserved in 10% buffered formalin. The brains in the second set (n = 6) were each divided into two portions. The first portion of tissue underwent homogenization with phosphate buffer (pH = 7.4) to create a 10% homogenate (*w*/*v*), which was then centrifuged at 4000 rpm for 15 min at 4 °C. The resulting supernatant was preserved at −20 °C for subsequent biochemical investigations [61]. Meanwhile, the second portion was kept at −80 °C for quantitative real-time PCR (RT-PCR) analysis.

### 2.7. Biochemical Assessments

#### 2.7.1. Colormetric Assays in the Brain

The acetylcholinesterase (ACHE) activities in the brain tissue homogenate was estimated by a commercial kit (Sigma-Aldrich, St. Louis, MO, USA (MAK119), following the standard protocol instructions. Additionally, the measurements of the lipid peroxidation product, malondialdehyde (MDA) (Cat. No. MD 2528), the activity of superoxide dismutase (SOD) (Cat. No. SD 2520), and the levels of total antioxidant capacity (TAC) (Cat. No. TA 2512) in brain tissue homogenates were estimated by commercially available kits (Biodiagnostic^®^, Inc., Giza, Egypt) following the manufacturer’s instructions.

#### 2.7.2. Fluorometric Assays in the Brain

The concentrations of brain monoamines, namely dopamine (DA), norepinephrine (NE), and serotonin (5-HT), were measured without delay using fluorometric kits that were commercially available (Sigma-Aldrich Co., St. Louis, MO, USA). As previously mentioned, the fluorometric method was employed to detect the monoamines present in the tissue samples, with specific excitation/emission wavelengths set at 320/480 nm for DA, 380/480 nm for NE, and 355/470 nm for 5-HT [62].

#### 2.7.3. Enzyme Linked Immunosorbent Assay (ELISA) in Brain Tissues

ELISA technique was employed to quantify inflammatory markers in brain tissue homogenates by ELISA kits (My BioSource, Inc., San Diego, CA, USA) according to the manufacturer’s recommendations. The levels of IL-1β (Cat.#MBS2023030), prostaglandin E2 (PGE-2) (Cat.#MBS262150), inducible nitric oxide synthase (iNOS) (Cat.#MBS723326), IL-6 (Cat.#MBS269892), cyclooxygenase-2 (COX-2) (Cat.#MBS725633) and TNF-α (Cat.#MBS175904). Moreover, the level of BDNF was also measured using a commercial kit (Cat.#MBS703433) provided by the same vendor.

#### 2.7.4. Gene Expression Measurement by Quantitative Real-Time Polymerase Chain Reaction (qRT-PCR)

The total RNA was isolated from brain tissue samples by the standard protocol provided by Qiagen (Germantown, MD, USA), per the manufacturer’s instructions. The concentration of mRNA was evaluated by a NanoDrop Spectrophotometer (Thermo Fisher Scientific, Waltham, MA, USA). Subsequently, 1 μg of mRNA was reverse transcribed into cDNA employing the PrimeScript^TM^ RT reagent Kit (Takara, Japan). The qRT-PCR were performed with SYBR Green reagents (Takara, Japan) on an Applied Biosystems instrument (StepOneTM, Waltham, MA, USA) with software version 3.1. The transcripts of B-cell lymphoma 2 protein (Bcl-2), Bcl-2-associated X protein (Bax), caspase-3, caspase-1, NLRP3, NF-κB, HO-1, Nrf2, TLR4, GSK-3β, PERK, CHOP, beclin-1, GRP78, PI3K, TrkB, AMPK, SIRT-1, AKT, CREB, HMGB1, RAGE, JAK-2 and STAT-3, were normalized to the expression level of the reference gene GAPDH. The relative expression of the target genes was evaluated by the Livak and Schmittgen method [63] with the formula: 2^−∆∆CT^. The list of forward and reverse primer sequences for each gene used in qPCR amplification is provided in Table 1.

### 2.8. Histopathological Evaluations

Formalin-fixed paraffin-embedded blocks of brain tissues were cut into 5 m-thick slices, stained using the usual H&E method, allowed to air dry, then observed under a light microscope and photographed at a magnification of 40× [64,65].

### 2.9. Statistical Analysis

Mean and SEM are used to describe the data. One-way ANOVA was used for the statistical analysis, and significance was determined at *p* <0.05. Tukey’s multiple comparisons test was then performed. The statistical analysis and data visualization were done using GraphPad Prism^®^ (Version 5, ISI, San Diego, CA, USA).

## 3. Results

In our study, the influences of PUN were examined on a normal control group. However, it did not display any substantial differences from the control group regarding the measured parameters and histopathological findings. Therefore, these results were not presented to avoid complicating the data.

### 3.1. Impacts of PUN on Motor Functions in Open Field Test in SI and/or MnCl_2_ Intoxicated Rats

In Figure 1, it is evident that SI or MnCl_2_ groups exhibited a significant intensification in latency time (by 3.9 and 4.9-fold, respectively) (Figure 1A), accompanied by a marked decrease in rearing (by 55.8% and 74.7%, respectively) (Figure 1B), grooming (by 53.2% and 68.1%, respectively) (Figure 1C), and ambulation frequencies (by 54.3% and 68.6%, respectively) (Figure 1D), compared to normal control group. Furthermore, rats exposed to SI + MnCl_2_ exhibited a more pronounced elevation in latency time and a decline in rearing, grooming, and ambulation frequencies compared to either SI or MnCl_2_ alone (Figure 1). In contrast, the administration of PUN to the rats exposed to SI, MnCl_2_, or SI with MnCl_2_ displayed a significant diminish in latency time (by 67.9%, 55.9%, and 48.8%, respectively), along with a remarkable increase in rearing (by 1.8, 2.7, and 4.8-fold, respectively), grooming (by 2.1, 2.7, and 3.9-fold, respectively), and ambulation frequencies (by 1.9, 2.4, and 4.2-fold, respectively) when linked to their corresponding untreated groups, SI, MnCl_2_, or SI with MnCl_2_, correspondingly. Of note, the administration of PUN to the SI rats restored the latency time and grooming frequency to normal values.

### 3.2. Impacts of PUN on Behavioural Parameters in Forced Swim Test (FST) and Y-Maze Test in SI and/or MnCl_2_-Intoxicated Rats

As displayed in Figure 2, SI or MnCl_2_ resulted in a meaningful rise in the immobility score (by 6.5 and 7.9-fold, respectively) (Figure 2A) and climbing score (by 2.7 and 3.1-fold, respectively) (Figure 2B), along with a noticeable decrease in the swimming score (by 73.4% and 82.1%, respectively) (Figure 2C) and percent of spontaneous alteration (by 31.6% and 41.4%, respectively) (Figure 2D), related to the normal control group. Notably, the concomitant SI + MnCl_2_ displayed even higher increases in immobility and climbing scores and a further decrease in swimming scores and percent of spontaneous alteration compared to either SI or MnCl_2_ alone. On the contrary, the administration of PUN to rats exposed to SI, MnCl_2_, or SI with MnCl_2_ led to a significant reduction in both immobility scores (by 70.9%, 61.6%, and 58.6%, respectively) and climbing scores (by 56.2%, 51.5%, and 52.3%, respectively), together with a notable increase in the swimming score (by 2.7, 3.1, and 4.7-fold, respectively) and percentage of spontaneous alteration (by 1.4, 1.5, and 1.6-fold, individually) related to the matching untreated groups, SI, MnCl_2_, or SI with MnCl_2_, respectively (Figure 2). Importantly, the administration of PUN to SI rats restored the climbing score in the FST to normal values.

### 3.3. Impacts of PUN on Brain Monoamine Levels and ACHE Activity in SI and/or MnCl_2_-Intoxicated Rats

Figure 3 illustrated that SI or MnCl_2_ resulted in a significant reduction in brain monoamines; specifically, there was a decrease by 25.7% and 67.2% in DA levels (Figure 3A), a decrease by 13.1% and 69.5% in NE levels (Figure 3B), and by 52.8% and 81.4% in 5-HT levels (Figure 3C), along with a meaningful elevation in ACHE activity by 4.3 and 5.6-fold, respectively (Figure 3D), relative to the normal control group. Notably, rats exposed to SI + MnCl_2_ exhibited a more pronounced decrease in brain monoamine contents and an increase in ACHE activity compared to rats exposed to individual SI or MnCl_2_. In contrast, the administration of PUN to rats exposed to SI, MnCl_2_, or SI with MnCl_2_ produced a marked increase in brain monoamine contents, specifically, an elevation by 27.2%, 132.9%, and 286.2% in DA levels; increase by 10.7%, 148.5%, and 61% in NE levels; and increase by 90.4%, 101.1%, and 197.9% in 5-HT levels, respectively (Figure 3A–C), together with a significant depletion in ACHE activity by 64.9%, 58.9%, and 61.4%, respectively, likened to their equivalent untreated groups, SI, MnCl_2_, or SI with MnCl_2_ (Figure 3D).

### 3.4. Impacts of PUN on Nrf2/HO-1 Pathway and Brain Oxidative Stress Biomarkers in SI and/or MnCl_2_-Intoxicated Rats

Figure 4 demonstrated a meaningful decline in the mRNA expression levels of both Nrf2 (by 48.9% and 59.9%, respectively) (Figure 4A) and HO-1 (by 54.6% and 63.2%, respectively) (Figure 4B); decreased SOD activity (by 68.3% and 80.3%, respectively) (Figure 4E) and TAC levels (by 44.9% and 64.5%, respectively) (Figure 4F), along with significant increase observed in brain MDA levels (by 7.2 and 11.6-fold, respectively) (Figure 4C) and iNOS levels (by 11.3 and 17.8-fold, respectively) (Figure 4D) following exposure to SI or MnCl_2_, related to the normal control group. Additionally, SI + MnCl_2_ rats displayed even greater depletion in brain MDA and iNOS levels, along with increased mRNA expression levels of both Nrf2 and HO-1, besides enhanced SOD activity and TAC levels compared to rats exposed to either SI or MnCl_2_ alone. In contrast, the administration of PUN to rats exposed to SI, MnCl_2_, or SI with MnCl_2_ leading to a considerable intensification in mRNA expression levels of both Nrf2 (by 62.8%, 81.3%, and 302.7%, respectively) and HO-1 (by 77.8%, 88.7%, and 408.8%, respectively); elevated SOD activity (by 2.9, 3, and 7-fold, respectively) and TAC levels (by 53.7%, 96.8%, and 130.5%, respectively), together with a significant diminution in brain MDA levels (by 73.5%, 67.9%, and 80.2%, respectively) and iNOS levels (by 79.6%, 66.6%, and 70.8%, respectively), related to the matching untreated groups, SI, MnCl_2_, or SI + MnCl_2_. It is worth noting that the administration of PUN to SI rats restored the brain iNOS level back to normal values.

### 3.5. Impacts of PUN on Brain Inflammatory Biomarkers (NF-ᴋB, TNF-α, IL-1β, IL-6, COX-2 and PGE2) in SI and/or MnCl_2_-Intoxicated Rats

As displayed in Figure 5, SI or MnCl_2_ substantially raised the mRNA expression level of NF-ᴋB (by 8.3 and 9.9-fold, respectively) (Figure 5A); the brain TNF-α levels by 5.5 and 9.6-fold; IL-1β by 5.6 and 8-fold; IL-6 by 8.5 and 11.9-fold; COX-2 by 2.9 and 4.5-fold, and PGE2 by 2.2 and 3.4-fold, correspondingly, related to the normal control group (Figure 5B–F). Furthermore, rats exposed to SI + MnCl_2_ exhibited an intensified inflammatory response, as evidenced by higher mRNA expression levels of NF-ᴋB, brain TNF-α, IL-1β, IL-6, COX-2, and PGE2 compared to rats exposed to either SI or MnCl_2_ alone. In contrast, administration of PUN to SI, MnCl_2_, or SI with MnCl_2_ substantially diminish the mRNA expression levels of NF-ᴋB (by 65.5%, 58.9%, and 53.1%, respectively), brain TNF-α (by 68.5%, 63.4%, and 72.8%, respectively), IL-1β (by 80.5%, 71.9%, and 59.1%, respectively), IL-6 (by 83.7%, 69.5%, and 59.2%, respectively), COX-2 (by 63.5%, 59.9%, and 68.9%, respectively), and PGE2 (by 40.9%, 52.9%, and 58.1%, respectively) related to their untreated equivalent groups (SI, MnCl_2_, or SI with MnCl_2_). It is worth noting that PUN treatment in SI rats restored the levels of brain IL-1β, IL-6, and COX-2 back to normal values.

### 3.6. Impacts of PUN on HMGB1/RAGE/TLR4; NLRP3/Caspase-1 and JAK-2/STAT-3 Pathways in SI and/or MnCl_2_-Intoxicated Rats

As depicted in Figure 6, SI or MnCl_2_ meaningfully upregulated the mRNA expression levels of TLR4 (by 9.9 and 11.3-fold, respectively) (Figure 6A), HMGB1 (by 5.5 and 7.8-fold, respectively) (Figure 6B), RAGE (by 5.9 and 7.6-fold, respectively) (Figure 6C), NLRP3 (by 7.8 and 9.8-fold, respectively) (Figure 6D), caspase-1 (by 7.2 and 8.5-fold, respectively) (Figure 6E), JAK-2 (by 10.2 and 11.8-fold, respectively) (Figure 6E), and STAT-3 (by 6.3 and 9.1-fold, respectively) (Figure 6F) likened to the normal control group. Additionally, rodents treated with SI + MnCl_2_ exhibited a more pronounced upregulation in mRNA expression levels of TLR4, HMGB1, RAGE, NLRP3, caspase-1, JAK-2, and STAT-3, compared to rats exposed to either SI or MnCl_2_ alone. In contrast, PUN treatment for SI, MnCl_2_, or SI with MnCl_2_ significantly downregulated the mRNA expression levels of TLR4 (by 89.7%, 48.5%, and 45.9%, respectively), HMGB1 (by 64.9%, 61.5%, and 58.4%, respectively), RAGE (by 61.5%, 51.3%, and 52.3%, respectively), NLRP3 (by 70.6%, 66.2%, and 65.3%, respectively), caspase-1 (by 62.9%, 52.5%, and 49.7%, respectively), JAK-2 (by 89.7%, 51.9%, and 48.7%, respectively), and STAT-3 (by 83.5%, 64.1%, and 55.5%, respectively) relative to their untreated corresponding groups (SI, MnCl_2_, or SI with MnCl_2_). Importantly, PUN treatment in SI rats restored the mRNA expression levels of TLR4, JAK-2, and STAT-3 back to normal values.

### 3.7. Impacts of PUN on PI3K/AKT/GSK-3β/CREB Pathway in SI and/or MnCl_2_-Intoxicated Rats

As clarified in Figure 7, SI or MnCl_2_ elicited substantial drop in the mRNA expression levels of PI3K (by 48.1% and 62.5%, respectively) (Figure 7A), AKT (by 49.8% and 68.1%, respectively) (Figure 7B), CREB (by 56.4% and 71.9%, respectively) (Figure 7D), TrKB (by 60.9% and 71.9%, respectively) (Figure 7E), and BDNF (by 58.1% and 64.5%, respectively) (Figure 7F), along with substantial enhancement in the mRNA expression level of GSK-3β (by 8.3 and 9.9-fold, respectively) (Figure 7C) linked to the normal control group. Moreover, rats exposed concurrently to SI + MnCl_2_ exhibited a more pronounced reduction in mRNA expression levels of PI3K, AKT, CREB, TrKB, and BDNF, along with an intensification in mRNA expression level of GSK-3β, likened to rats exposed to either SI or MnCl_2_ alone. However, SI, MnCl_2_, or SI with MnCl_2_ groups received PUN displayed a marked enhancement in the mRNA expression levels of PI3K (by 1.7, 2.1, and 3.8-fold, respectively), AKT (by 1.8, 2.4, and 4.6-fold, respectively), CREB (by 2.1, 2.6, and 5.2-fold, respectively), TrKB (by 2.2, 2.4, and 4.2-fold, respectively), and BDNF (by 1.83, 1.8, and 2.4-fold, respectively), together with a marked reduction in mRNA expression levels of GSK-3β (by 72.4%, 62.2%, and 45.6%, respectively) when it was likened to their untreated corresponding groups (SI, MnCl_2_, or SI with MnCl_2_). Remarkably, PUN treatment in SI rats restored the mRNA expression level of AKT back to normal values.

### 3.8. Impacts of PUN on Endoplasmic Reticulum Stress Biomarkers (PERK, GRP78 and CHOP) and Apoptotic Biomarkers (Bcl-2, Bax and Caspase-3) in SI and/or MnCl_2_-Intoxicated Rats

As depicted in Figure 8, SI or MnCl_2_ had shown mRNA upregulation of PERK (by 7.5 and 9.3-fold, respectively) (Figure 8A), GRP78 (by 8.2 and 9.9-fold, respectively) (Figure 8B), CHOP (by 8.9 and 10.5-fold, respectively) (Figure 8C), Bax (by 8 and 9.4-folds, respectively) (Figure 8E) and caspase-3 (by 7.9 and 9.3-folds, respectively) (Figure 8F), along with mRNA downregulation of Bcl-2 (by 44.7% and 59.9%, respectively) (Figure 8D), compared to the normal control group. Furthermore, SI + MnCl_2_ rats exhibited a noticeable elevation in mRNA expression levels of PERK, GRP78, CHOP, Bax, and caspase-3, along with a decrement in mRNA expression level of Bcl-2, related to individual SI or MnCl_2_ groups. Conversely, PUN administration to SI, MnCl_2_, or SI with MnCl_2_ caused a noteworthy downregulation in mRNA expression levels of PERK (by 65.8%, 57.8%, and 50.6%, respectively), GRP78 (by 70.6%, 61.4%, and 56.8%, respectively), CHOP (by 75%, 64.6%, and 53.6%, respectively), Bax (by 66.7%, 53.7% and 45.6%, respectively) and caspase-3 (by 68.3%, 53.1% and 50%, respectively), together with a substantial upregulation in mRNA expression level of Bcl-2 (by 1.7, 1.9 and 3.9-folds, respectively), matched to their untreated equivalent groups (SI, MnCl_2_, or SI with MnCl_2_).

### 3.9. Impacts of Punicalagin on AMPK/SIRT-1 Pathway in SI and/or MnCl_2_-Intoxicated Rats

As illustrated in Figure 9, SI or MnCl_2_ resulted in a substantial diminution in mRNA expression levels of AMPK (by 44.3% and 61.1%, respectively) (Figure 9A), SIRT-1 (by 50.3% and 62.3%, respectively) (Figure 9B), and beclin-1 (by 53.2% and 69.2%, respectively) (Figure 9C), compared to the normal control group. Interestingly, SI + MnCl_2_ led to a more pronounced reduction in mRNA expression levels of AMPK, SIRT-1, and beclin-1 relative to rats exposed to either SI or MnCl_2_ alone. Conversely, PUN treatment for SI, MnCl_2_, or SI with MnCl_2_ resulted in a remarkable upregulation in mRNA expression levels of AMPK (by 1.7, 2, and 3.1-fold, respectively), SIRT-1 (by 1.8, 2.1, and 3.9-fold, respectively), and beclin-1 (by 1.9, 2.5, and 3.9-fold, respectively) matched to their untreated equivalent groups (SI, MnCl_2_, or SI with MnCl_2_). It is worth noting that PUN in SI rats restored the mRNA expression level of AMPK back to its normal values.

### 3.10. Impacts of Punicalagin on Histopathological Alterations of Different Brain Regions in SI and/or MnCl_2_-Intoxicated Rats

As illustrated in Figure 10, brain sections containing the cerebral cortex, subiculum, fascia dentata in the hippocampus, striatum, and substantia nigra areas from the normal control group, exhibited normal histological architecture (Inserts a1–5). However, in the isolated group, nuclear pyknosis and degeneration were observed in most neurons of both the cerebral cortex (Insert b1) and fascia dentata (Insert b3), while no histopathological changes were noted in the subiculum, striatal areas, and substantia nigra (Inserts b2–5). Alternatively, the Mn group displayed a marked nuclear pyknosis and degeneration in the cerebral cortex (Insert c1), subiculum (Insert c2), and fascia dentate (Insert c3). In addition, the striatal areas exhibited the creation of large-sized focal eosinophilic plaques (Insert c4), and some neurons of the substantia nigra displayed degeneration (Insert c5). Interestingly, in the Mn Isolated group, severe nuclear pyknosis and degeneration were observed in most neurons of the cerebral cortex (Insert d1), fascia dentata (Insert d3), and substantia nigra (Insert d5), along with all neurons of the subiculum (Insert d2). Furthermore, the striatal areas of this group exhibited marked establishment of multiple large-sized focal eosinophilic plaques with the loss of neurons (Insert d4). In contrast, treatment with PUN to SI, MnCl_2_, or SI with MnCl_2_ ameliorated all the aforementioned histopathological alterations in the explored brain regions by varying degrees, compared to their untreated corresponding groups (SI, MnCl_2_, or SI with MnCl2, respectively). It is worth noting that PUN treatment resulted in apparent normal histological architecture in all investigated brain areas in the SI + PUN group (Inserts e1–5), along with neurons of the cerebral cortex (Insert f1), hippocampus (subiculum and fascia dentate) (Inserts f2 and f3), and striatum (Insert f4) in the Mn + PUN group. Moreover, neurons of the cerebral cortex (Insert g1) and hippocampus (subiculum and fascia dentate) (Inserts g2 and g3) in the Mn Isolated + PUN group also exhibited normal histological architecture. However, there was still slight degeneration in some neurons of the substantia nigra in Mn + PUN group (Insert f5), along with a slight formation of focal fine plaques in the striatal area (Insert g4) and slight degeneration in some neurons of the substantia nigra noted in the Mn Isolated + PUN group (Insert g5).

## 4. Discussion

PD is a debilitating neurological condition marked by progressive loss of dopaminergic neurons and accumulation of neurotoxic α-synuclein. This leads to a deficiency of dopamine (DA), which is responsible for motor symptoms [66]. Notably, SI is a potent stressor that can cause behavioural changes and neurodegeneration in both humans and rodents, resulting in memory impairment, anxiety, and depression [67]. However, the underlying mechanisms by which SI may impact the progression of PD in affected individuals have not been well understood. Our study examined the interconnection between SI and the progression of PD induced experimentally by MnCl_2_ via comparing PD-mediated biochemical, behavioural, and histopathological alterations, as well as outlining the possible molecular mechanisms involved in this relationship. Of particular significance, plant-derived phytochemicals with antioxidant and anti-inflammatory activities have gained much popularity as a potential therapeutic approach against neurodegenerative diseases. This approach targets multiple pathways, and one compound that has attracted considerable attention is PUN. PUN has been revealed to own various health benefits, including antioxidant, anti-inflammatory, and neuroprotective activities [36,68]. Consequently, this investigation targeted to discover the beneficial properties of PUN on behavioural, biochemical, and histopathological modifications in SI rats with PD. Additionally, the study elucidated the molecular targets through which PUN exerts its beneficial effects in ameliorating PD progression induced by MnCl_2_ in SI rats, focusing on HMGB1/RAGE/TLR4/ NF-ᴋB, NLRP3/Caspase-1, JAK-2/STAT-3, PI3K/AKT/GSK-3β/CREB, AMPK/SIRT-1, Nrf2/HO-1, and PERK/CHOP/Bcl-2 cues.

Our results demonstrated that rats exposed to concurrent SI and MnCl_2_ intoxication exhibited more pronounced alterations in spatial working memory and motor activity. This was proven by a deterioration in cognitive function in the Y-maze test, increased latency in the OFT, decreased swimming score, and augmented immobility score in the FST. These observations suggest that the rats experienced negative mood, despair behaviour, bradykinesia, rigidity, motor deficits, and impaired learning and memory functions. Notably, these effects were more pronounced in the SI + MnCl_2_ group compared to the individual SI or MnCl_2_ groups. These findings suggested possible synergistic effects between SI and MnCl_2_ in worsening the behavioural outcomes of PD. The observed behavioural alterations in the SI + MnCl_2_ group can be attributed to significant depletion in the brain monoamines levels (dopamine, norepinephrine, and serotonin) and marked increment in ACHE activity, together with severe nuclear pyknosis and degeneration observed in the histopathological findings of the cerebral cortex and hippocampus in the SI + MnCl_2_ group, related to individual SI or PD groups. It is worth noting that these ongoing alterations could be partly attributed to enhanced oxidative stress, neuroinflammation, and neuronal damage in the brains of SI + MnCl_2_ rats. Disturbances in cholinergic neurotransmission, as indicated by the substantial rise in ACHE activity in the brains of the SI + MnCl_2_ group, may have a central role in the impaired learning, memory, and locomotor functions observed in this group, where maintaining normal cholinergic neurotransmission system is essential for normal brain functions, particularly locomotor and memory functions. Our results align with the findings of Abu-Elfotuh et al. [69], where exposure to MnCl_2_ resulted in hippocampal neuronal cell death and a decline in cognitive, motor, and memory functions, as well as monoamine levels and ACHE activity in rats.

Similarly, the study by Famitafreshi et al. supported our findings, demonstrating the adverse effects of SI on the hippocampus and cerebral cortex, which have a key role in learning, memory, and cognitive functions, as well as dopamine levels [70,71]. On the other hand, treatment with PUN in the SI and/or MnCl_2_-intoxicated groups significantly ameliorated all the pervious alternations, compared to their corresponding control groups (SI, MnCl_2_, or SI with MnCl_2_). These impacts could be endorsed to PUN’s antioxidant and anti-inflammatory actions [46,47,48,49,50].

It is interesting to note that PD development involves multiple interconnected processes, for instance, oxidative stress, neuroinflammation, protein aggregation, ER stress, dysregulated autophagy, and apoptosis of dopaminergic neurons [10,26,34,72,73,74]. In our study, rats exposed to SI and MnCl_2_ intoxication experienced significant disruption in their redox balance. This was evident through the depletion of antioxidant defense mechanisms, including reduced expression of Nrf2 and HO-1 genes, decreased levels of SOD and TAC, and increased formation of ROS and reactive nitrogen species (RNS), as indicated by elevated iNOS levels and MDA compared to the individual SI or MnCl_2_ groups. These findings suggested an augmentation between SI and MnCl_2_ in promoting ROS production and impairing antioxidant defenses. The observed disturbances in redox balance can be attributed to the upregulation of GSK-3β expression in SI, MnCl_2_, or SI + MnCl_2_ groups. Enhanced GSK-3β expression can inhibit Nrf2 activation, leading to the downregulation of antioxidant genes including HO-1 and SOD [75]. This influence was particularly noted in the SI + MnCl_2_ group compared to the respected individual groups. In contrast, treatment with PUN in SI and/or MnCl_2_-intoxicated groups effectively mitigated these redox imbalances compared to their respective control groups (SI, MnCl_2_, or SI with MnCl_2_) because of antioxidant properties of PUN [37]. Moreover, PUN treatment led to the downregulation of GSK-3β expression, with subsequent stimulation of Nrf2/HO-1 pathway and upregulation of its target genes in the SI, MnCl_2_, or SI + MnCl_2_ groups. Previous studies further supported our findings which had also demonstrated the notable antioxidant effects of PUN in various diseases [76,77].

Another significant pathophysiological event implicated in the progression of PD is the triggering of inflammatory signaling pathways, with a particular focus on the HMGB1 pathway. Increased expression of HMGB1 has been connected with the pathogenesis of PD [78,79]. The upregulation of HMGB1 triggers neuroinflammation by activating the NF-κB pathway. HMGB1 activates its receptors, RAGE, and TLR4, which synergistically act with elevated levels of ROS and increased expression of GSK-3β to induce NF-κB activation [80]. Once NF-κB is activated, it leads to the release of NLRP3 and pro-inflammatory mediators, including TNF-α, IL-1β, IL-6, COX-2, PGE-2, and iNOS [81], which was also observed in our study in the SI, MnCl_2_, and SI + MnCl_2_ groups. The initiation of NLRP3 further triggers caspase-1 activation, resulting in the cleavage of pro-IL-1β and the formation of mature IL-1β, thus strengthening the inflammatory response and promoting neuronal cell death [82,83], which supported our findings in SI, MnCl_2_, and SI + MnCl_2_ groups. Additionally, the elevation of pro-inflammatory mediators, primarily IL-6, observed in the SI, MnCl_2_, and SI + MnCl_2_ groups may induce JAK-2/STAT-3 pathway that involved in neuroinflammatory processes [84,85]. Upon activation, JAK-2 phosphorylates STAT-3, leading to its nuclear translocation and subsequent astroglial overactivation and neuronal damage [86,87], which aligns with our findings in the SI, MnCl_2_, and SI + MnCl_2_ groups. Interestingly, rats exposed to both SI and MnCl_2_ intoxication exhibited a more pronounced inflammatory response, as evidenced by increased gene expression of HMGB1, RAGE, TLR4, NF-κB, NLRP3, caspase-1, JAK-2, and STAT-3, along with elevated levels of TNF-α, IL-1β, IL-6, iNOS, COX-2, and PGE-2 compared to the SI or MnCl_2_ groups alone. These findings highlight the exacerbating effect of SI on neuroinflammation in rats with PD induced by MnCl_2_.

On the contrary, the administration of PUN substantially diminished the abnormal activation of the aforementioned proinflammatory and inflammatory pathways in the SI, MnCl_2_, and SI + MnCl_2_ groups compared to their respective control groups. These favourable consequences could be attributed to the anti-inflammatory properties of PUN [88]. Notably, PUN suppressed the upregulation of HMGB1 expression induced by SI, MnCl_2_, and SI + MnCl_2_, thereby inhibiting the activation of both RAGE and TLR4 receptors. Furthermore, PUN exhibited antioxidant actions and downregulated the expression of GSK-3β, resulting in the suppression of NF-κB activation. Consequently, NF-κB inhibition by PUN resulted in the suppression of NLRP3 release and the formation of pro-inflammatory mediators. This preventedNLRP3 activation and consequently blocked the activity of caspase-1, thereby reducing the maturation of IL-1β, in agreement with previous studies [89,90]. Additionally, PUN attenuated the expression of pro-inflammatory mediators, which contributed to the suppression of the JAK-2/STAT-3 signaling pathway and reduced astroglial activation. Besides its role as an Nrf2 activator and antioxidant, PUN-induced upregulation of Nrf2 exerted anti-inflammatory properties by hindering NF-κB activation, which is typically triggered by ROS, thereby reducing the release of pro-inflammatory mediators [91,92,93]. It is worth noting that the anti-inflammatory properties of PUN have been previously stated in various diseases, where it effectively suppressed the release of pro-inflammatory mediators [49,88,94].

Prior studies have established the association between the dysregulation of the PI3K/AKT signaling pathway and the development of PD [12,95]. In this context, rats exposed to SI, MnCl_2_ intoxication, or SI combined with MnCl_2_ exhibited a disruption in PI3K/AKT pathway. This disruption was characterized by reduced expressions of PI3K, AKT, CREB, and TrKB, as well as decreased BDNF levels, accompanied by an elevated expression of GSK-3β in these groups.

Conversely, administration of PUN to the SI, MnCl_2_, or SI + MnCl_2_ groups led to increased expressions of PI3K, AKT, CREB, and TrKB, along with downregulation of GSK-3β expression, in comparison to their corresponding control groups. These influences could be endorsed to the capability of PUN to reverse the repression of the PI3K/AKT pathway induced by SI, MnCl_2_, or SI + MnCl_2_. Specifically, the elevation of PI3K phosphorylated AKT, which in turn phosphorylated both GSK-3β and CREB, resulting in the stimulation of CREB and inhibition of GSK-3β activity. Activated CREB, in turn, resulted in the upregulation of BDNF, which interacted with TrKB to promote neuronal plasticity, memory regulation and inhibit neuronal apoptosis [49,96,97]. Furthermore, the binding of BDNF to TrKB stimulated the PI3K/AKT axis, reinforcing the activation of CREB and subsequent transcription of BDNF, establishing a reciprocal relationship between the PI3K/AKT and CREB/BDNF/TrKB pathways [98]. It is noteworthy that GSK-3β is accompanying to increase the oxidative stress, neuroinflammation, and dopaminergic neuronal apoptosis [73,99,100]. This can be justified by the capability of GSK-3β overexpression to negatively regulate Nrf2/HO-1 activation by promoting Nrf2 degradation while positively regulating NF-κB initiation by facilitating IκBα phosphorylation and NF-κB nuclear translocation [101,102]. Therefore, the observed GSK-3β overexpression might contribute to the ongoing oxidative stress, neuroinflammation, and neuronal apoptosis observed in the SI, MnCl_2_, or SI + MnCl_2_ groups.

Another mechanistic factor implicated in PD and regulating apoptosis is ER stress. Numerous studies have reported that the buildup of misfolded proteins leading to ER stress is a significant contributor to the development of PD, where excessive or prolonged ER stress had been closely associated with the initiation of neuronal apoptosis [14,17,103,104,105]. In the same vein, rats exposed to SI, MnCl_2_, or SI + MnCl_2_ intoxication exhibited sustained ER stress, as corroborated by an increase in ER stress markers, counting the gene expression of PERK, GRP78, and CHOP, relative to the normal control group. These observed changes can be attributed to the capability of SI, MnCl_2_, or SI + MnCl_2_ to induce the overproduction of ROS, which provoked the buildup of misfolded proteins, resulting in excessive ER stress in these groups. Specifically, α-syn aggregates-mediated sustained ER stress overactivated UPR that stimulated key sensing proteins, particularly; PERK, through the detachment of GRP78, consequently this severe/prolonged ER stress led to an elevation in the pro-apoptotic factor, CHOP, which initiated the apoptotic cascades, via enhancing caspases cascade and decreasing Bcl-2 [14,103,104]. In addition to ER stress-induced apoptosis, oxidative stress, and neuroinflammation also stimulated neuronal apoptosis, where enhanced ROS stimulate caspase-dependent apoptosis [19,106,107]. Besides, the elevation of GSK-3β expression induced apoptotic signals via downregulating Bcl-2 expression and upregulating Bax expression, together with its mitochondrial translocation, hence stimulating the release of cytochrome c into the cytosol that consequently triggers both caspase-9 and -3 to initiate dopaminergic neuronal apoptosis [108,109]. This was supported by the remarkable elevation in apoptotic markers, including the gene expression of Bax and caspase-3 content, alongside with marked downregulation in the gene expression of anti-apoptotic Bcl-2 in these three groups, compared to normal control group. Worth mentioning, SI concurrently with MnCl_2_ intoxication elicited pronounced elevation in ER stress markers and apoptotic markers, compared to that in either SI group or MnCl_2_ intoxicated group separately.

In contrast, PUN, by virtue of its depicted antioxidant properties, demonstrated the ability to mitigate SI and/or MnCl_2_-induced oxidative stress, which may suppress ROS-mediated abnormal α-syn protein misfolding, resulting in abating SI and/or MnCl_2_-mediated excessive ER stress markers alternations. In this frame, the depicted abilities of PUN to abrogate SI and/or MnCl_2_-mediated enhancement in ER stress, oxidative stress, neuroinflammation, and GSK-3β expression collectively resulted in halting SI and/or MnCl_2_-induced neuronal apoptosis and diminish the apoptosis biomarkers. These results align with previous studies that reported decrement of ER stress markers and apoptotic markers upon treatment with PUN in various diseases [50,110,111,112].

Apart from inducing neuronal apoptosis, sustained ER stress also contributed to impaired autophagy, leading to disruption of the balance between neuronal apoptosis and autophagy, which is crucial in maintaining homeostasis, and this imbalance, is principally convoluted in the pathogenesis of PD [20]. Indeed, autophagy plays a crucial role in clearing misfolded proteins, like α-syn, which may produce toxicity to dopaminergic neurons, resulting in dopaminergic neuronal cell death in PD [104,113]. Therefore, enhancing autophagy may be a potential therapeutic strategy for PD via clearance of α-syn misfolded proteins and improving neuronal survival [18,21,114]. Notably, AMPK/SIRT-1 pathway is a key regulator of autophagy [115,116], and its suppression had been observed in PD, leading to the accumulation of misfolded α-syn. This leads to increased dopaminergic neuronal death and locomotor abnormalities accompanying with PD [117]. In the context of SI, MnCl_2_, or SI + MnCl_2_ groups, the observed exaggerated ER stress resulted in enhanced neuronal apoptosis and alternation in autophagy and autophagy biomarkers compared to the normal control group. Significantly, SI + MnCl_2_ group displayed more noticeable impairment in autophagic mechanisms, specifically the prominent suppression of AMPK/SIRT-1/Beclin-1 pathway, compared to either SI or MnCl_2_ alone groups. In contrast, the administration of PUN to SI, MnCl_2_, or SI + MnCl_2_ groups demonstrated a significant improvement in autophagy. These effects could be attributed to PUN’s capability to reverse the repressive effects of SI, MnCl_2_, or SI + MnCl_2_ on the AMPK/SIRT-1 pathway. AMPK activation leads to SIRT-1 activation by increasing NAD/NADH ratio, promoting autophagy and upregulated the expression of the autophagy-related gene beclin-1, leading to enhanced clearance of aggregated misfolded α-syn and subsequently suppressing neuronal apoptosis while promoting neuronal survival and improving locomotor function in PD [118,119,120] Importantly, the clearance of accumulated misfolded α-syn through enhanced autophagy may alleviate ER stress and return ER homeostasis, resulting in the suppression of neuronal apoptosis in dopaminergic neurons [121]. Additionally, SIRT-1 can sequester the pro-apoptotic protein Bax within the cytoplasm, preserving dopaminergic neurons and inhibiting neuronal apoptosis [122]. It is worth noting that AMPK and SIRT-1 mutually stimulate each other’s activity, and there is a positive amplification loop between AMPK and AKT [123,124]. Earlier studies have documented the valuable properties of PUN in numerous disease states, partly by acting as an AMPK activator to enhance autophagy through the AMPK/SIRT-1 pathway [118,119].

Furthermore, it is vital to mention that histopathological findings validated all the aforementioned biochemical results of our study. The investigation of H&E-stained sections from the cerebral cortex, subiculum, fascia dentata, and hilus in the hippocampus, striatum, and substantia nigra regions of rats exposed to SI and/or MnCl_2_ intoxication, which revealed varying degrees of histopathological alterations. These changes included nuclear pyknosis and degeneration in most neurons of the cerebral cortex and fascia dentata in the SI, MnCl_2_, or SI + MnCl_2_ groups. The MnCl_2_-intoxicated group also showed degeneration of some neurons in the subiculum and substantia nigra, while the SI + MnCl_2_ group exhibited degeneration in all neurons of the substantia nigra and subiculum. Additionally, the striatal areas in the MnCl_2_ or SI + MnCl_2_ groups displayed the formation of large eosinophilic plaques, and in SI + MnCl_2_ group, there was a loss of neurons along with multiple large eosinophilic plaques.

On the other hand, PUN treatment markedly relieved the SI and/or MnCl_2_-induced histopathological alterations in all examined brain regions to varying degrees. It also preserved the normal histological picture in the cerebral cortex, hippocampus (subiculum and fascia dentata), and striatum in the SI + PUN group, as well as in the neurons of the cerebral cortex, hippocampus, and striatum in Mn + PUN group. Of note, there were slight degenerative changes in some neurons of the substantia nigra in both Mn + PUN and Mn Isolated + PUN groups, along with the slight formation of focal fine plaques in the striatal area in the Mn Isolated + PUN group.

## 5. Conclusions

This investigation studied the impact of SI on the progression of PD by establishing a model of PD with concurrent exposure to SI. The study explored the interconnection between SI and PD and examined its effects on behavioural, biochemical, and histopathological findings; together with outlining the possible molecular contributors underlying this relationship.

Co-exposure to SI and MnCl_2_ revealed that SI, as a psychological stressor, along with its associated depression, accelerated the progression of PD and worsened its prognosis. The behavioural, neurochemical, and histopathological alterations were more pronounced in the group exposed to SI with MnCl_2_ compared to the group exposed to MnCl_2_ alone. SI exacerbated oxidative stress, neuroinflammation, ER stress, and neuronal apoptosis while reducing autophagy in PD rats, compared to either SI or PD solo groups.

It is worth noting that PUN exhibited promising neuroprotective effects against SI and/or MnCl_2_ intoxication in rats. It exerted antioxidant, anti-inflammatory, and anti-apoptotic properties, improved behavioural disturbances, normalized neurotransmitter levels, and mitigated histopathological changes. Mechanistically, PUN alleviated the progression of PD in rats exposed to SI with MnCl_2_ primarily by modulating HMGB1/RAGE/TLR4/NF-κB/NLRP3/Caspase-1, JAK-2/STAT-3, PI3K/AKT/GSK-3β/CREB, AMPK/SIRT-1, Nrf2/HO-1, and PERK/CHOP/Bcl-2 pathways (as shown in Graphical abstract).

Our study provided new evidence supporting using PUN as a safe, natural, and multitarget neuroprotective approach for halting neuronal damage, neurodegeneration, and enhancing neuronal survival, improving motor functions, and prognosis of PD. However, more pharmacological studies are needed before initiating clinical trials.

## Figures and Tables

**Figure 1 pharmaceutics-15-02420-f001:**
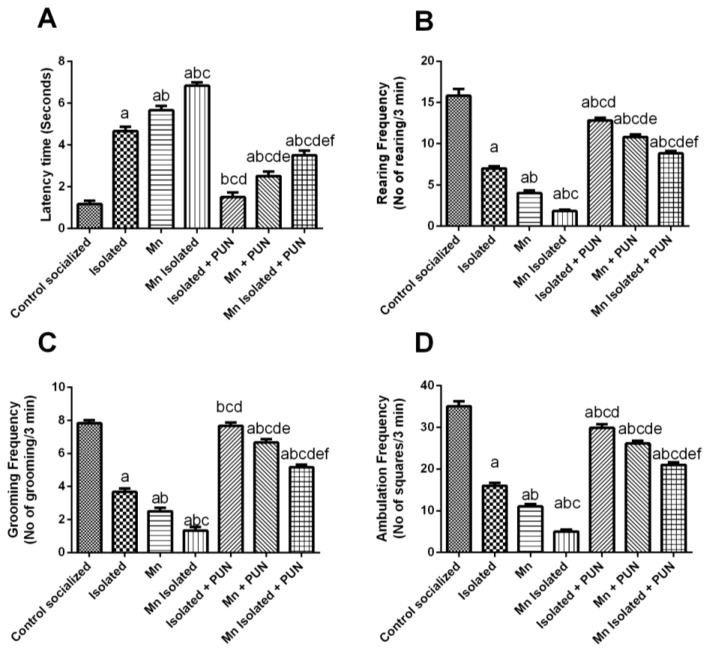
Impacts of PUN on motor functions in open field test in SI and/or MnCl_2_-intoxicated rats. (**A**) Latency time, (**B**) Rearing frequency, (**C**) Grooming frequency, and (**D**) Ambulation frequency. Control socialized: Normal control socialized rats, 4 per cage and fed rodent chow diet, Isolated: Rats were placed individually in laboratory cages, Mn: Rodents were treated daily with MnCl_2_ (10 mg/kg, i.p.) to induce PD, Mn Isolated: Socially isolated rats received MnCl_2_, Isolated + PUN: Socially isolated rats received PUN (30 mg/kg, p.o.) for 5 weeks, Mn + PUN: Rats received MnCl_2_ and PUN, Mn Isolated + PUN: Socially isolated rats received MnCl_2_ and PUN. Data are exhibited as mean (n = 6) ± S.E.M. Significant difference was calculated relative to ^(a)^ Normal control, ^(b)^ Isolated, ^(c)^ MnCl_2_, ^(d)^ Mn + Isolated, ^(e)^ Isolated + PUN, and ^(f)^ Mn + PUN groups. The significance level at *p*-value of <0.05.

**Figure 2 pharmaceutics-15-02420-f002:**
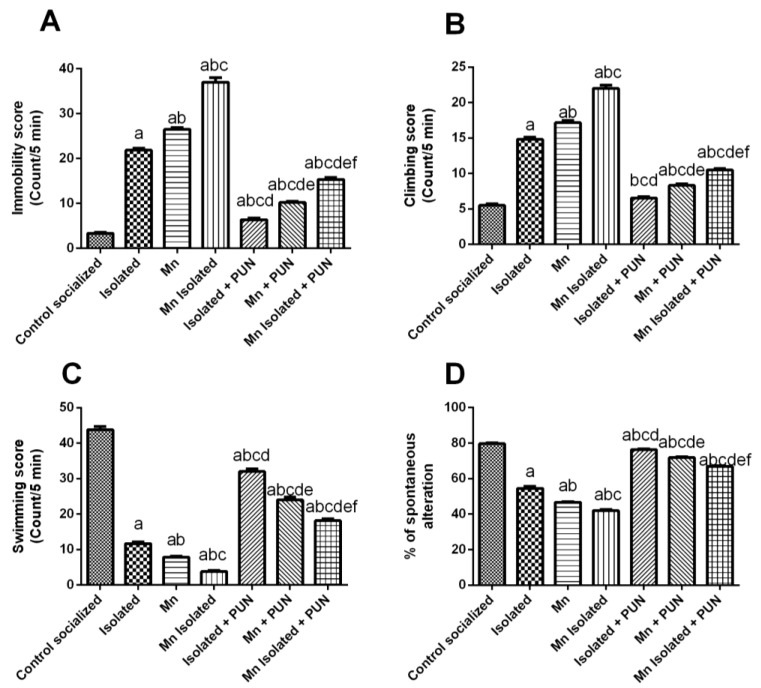
Impacts of PUN on motor functions on behavioural parameters in FST and Y-maze test in SI and/or MnCl_2_-intoxicated rats. (**A**) Immobility score, (**B**) Climbing score, (**C**) Swimming score, and (**D**) Percentage of spontaneous alteration. Control socialized: Normal control socialized rats, 4 per cage and fed rodent chow diet, Isolated: Rats were placed individually in laboratory cages, Mn: Rats were treated daily with MnCl_2_ (10 mg/kg, i.p.) to induce PD, Mn Isolated: Socially isolated rats received MnCl_2_, Isolated + PUN: Socially isolated rats received PUN (30 mg/kg, p.o.) for 5 weeks, Mn + PUN: Rats received MnCl_2_ and PUN, Mn Isolated + PUN: Socially isolated rats received MnCl_2_ and PUN. Data are exhibited as mean (n = 6) ± S.E.M. Significant difference was calculated relative to ^(a)^ Normal control, ^(b)^ Isolated, ^(c)^ MnCl_2_, ^(d)^ Mn+ Isolated, ^(e)^ Isolated + PUN, and ^(f)^ Mn + PUN groups. The significance level at *p*-value of <0.05.

**Figure 3 pharmaceutics-15-02420-f003:**
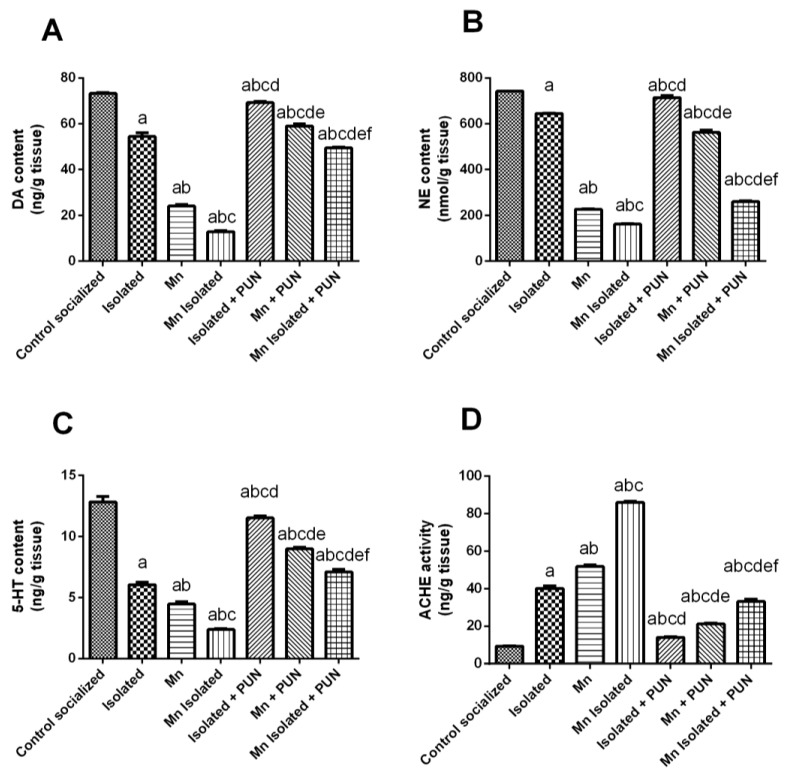
Impacts of PUN on brain monoamines contents and ACHE activity in SI and/or MnCl_2_-intoxicated rats. (**A**) Dopamine, (**B**) Norepinephrine, (**C**) Serotonin (5-HT) levels, and (**D**) ACHE activity. Control socialized: Normal control socialized rats, 4 per cage and fed rodent chow diet, Isolated: Rats were placed individually in laboratory cages, Mn: Rodenrts were treated daily with MnCl_2_ (10 mg/kg, i.p.) to induce PD, Mn Isolated: Socially isolated rats received MnCl_2_, Isolated + PUN: Socially isolated rats received PUN (30 mg/kg, p.o.) for 5 weeks, Mn + PUN: Rats received MnCl_2_ and PUN, Mn Isolated + PUN: Socially isolated rats received MnCl_2_ and PUN. Data are exhibited as mean (n = 6) ± S.E.M. Significant difference was calculated relative to ^(a)^ Normal control, ^(b)^ Isolated, ^(c)^ MnCl_2_, ^(d)^ Mn+ Isolated, ^(e)^ Isolated + PUN, and ^(f)^ Mn + PUN groups. The significance level at *p*-value of <0.05.

**Figure 4 pharmaceutics-15-02420-f004:**
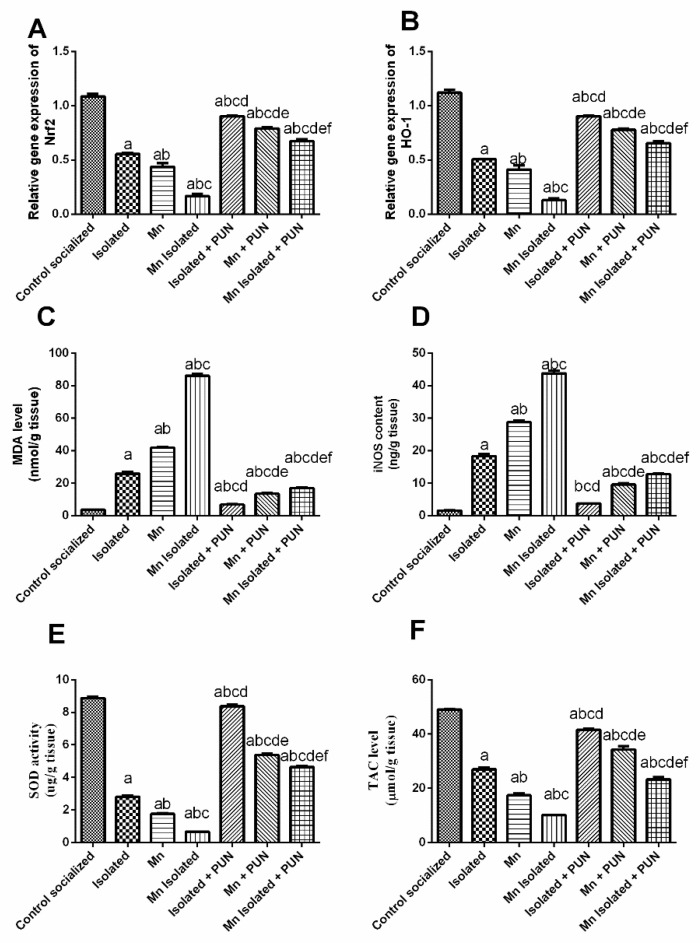
Impacts of PUN on Nrf2/HO-1 pathway and brain oxidative stress biomarkers in SI and/or MnCl_2_-intoxicated rats. (**A**) mRNA expression level of Nrf2 and (**B**) mRNA expression level of HO-1, (**C**) MDA, (**D**) iNOS, (**E**) SOD, and (**F**) TAC levels. Control socialized: Normal control socialized rats, 4 per cage and were fed rodent chow diet, Isolated: Rats were placed individually in laboratory cages, Mn: Rodents were treated daily with MnCl_2_ (10 mg/kg, i.p.) to induce PD, Mn Isolated: Socially isolated rats received MnCl_2_, Isolated + PUN: Socially isolated rats received PUN (30 mg/kg, p.o.) for 5 weeks, Mn + PUN: Rats received MnCl_2_ and PUN, Mn Isolated + PUN: Socially isolated rats received MnCl_2_ and PUN. Data are exhibited as mean (n = 6) ± S.E.M. Significant difference was calculated relative to ^(a)^ Normal control, ^(b)^ Isolated, ^(c)^ MnCl_2_, ^(d)^ Mn+ Isolated, ^(e)^ Isolated + PUN, and ^(f)^ Mn + PUN groups. The significance level at *p*-value of <0.05.

**Figure 5 pharmaceutics-15-02420-f005:**
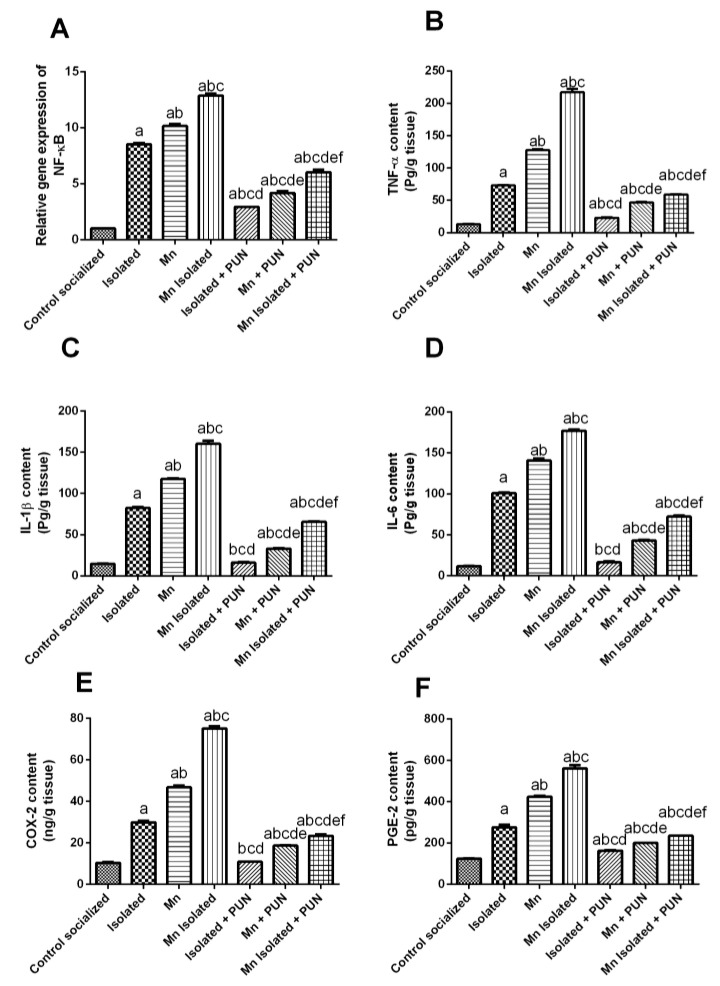
Impacts of PUN on brain inflammatory biomarkers in SI and/or MnCl_2_-intoxicated rats. (**A**) mRNA expression level of NF-ᴋB, (**B**) TNF-α level, (**C**) IL-1β level, (**D**) IL-6 level, (**E**) COX-2 level, and (**F**) PGE-2 level. Control socialized: Normal control socialized rats, 4 per cage and fed rodent chow diet, Isolated: Rats were placed individually in laboratory cages, Mn: Rats were treated daily with MnCl_2_ (10 mg/kg, i.p.) to induce PD, Mn Isolated: Socially isolated rats received MnCl_2_, Isolated + PUN: Socially isolated rats received PUN (30 mg/kg, p.o.) for 5 weeks, Mn + PUN: Rats received MnCl_2_ and PUN, Mn Isolated + PUN: Socially isolated rats received MnCl_2_ and PUN. Data are exhibited as mean (n = 6) ± S.E.M. Significant difference was calculated relative to ^(a)^ Normal control, ^(b)^ Isolated, ^(c)^ MnCl_2_, ^(d)^ Mn+ Isolated, ^(e)^ Isolated + PUN, and ^(f)^ Mn + PUN groups. The significance level at *p*-value of <0.05.

**Figure 6 pharmaceutics-15-02420-f006:**
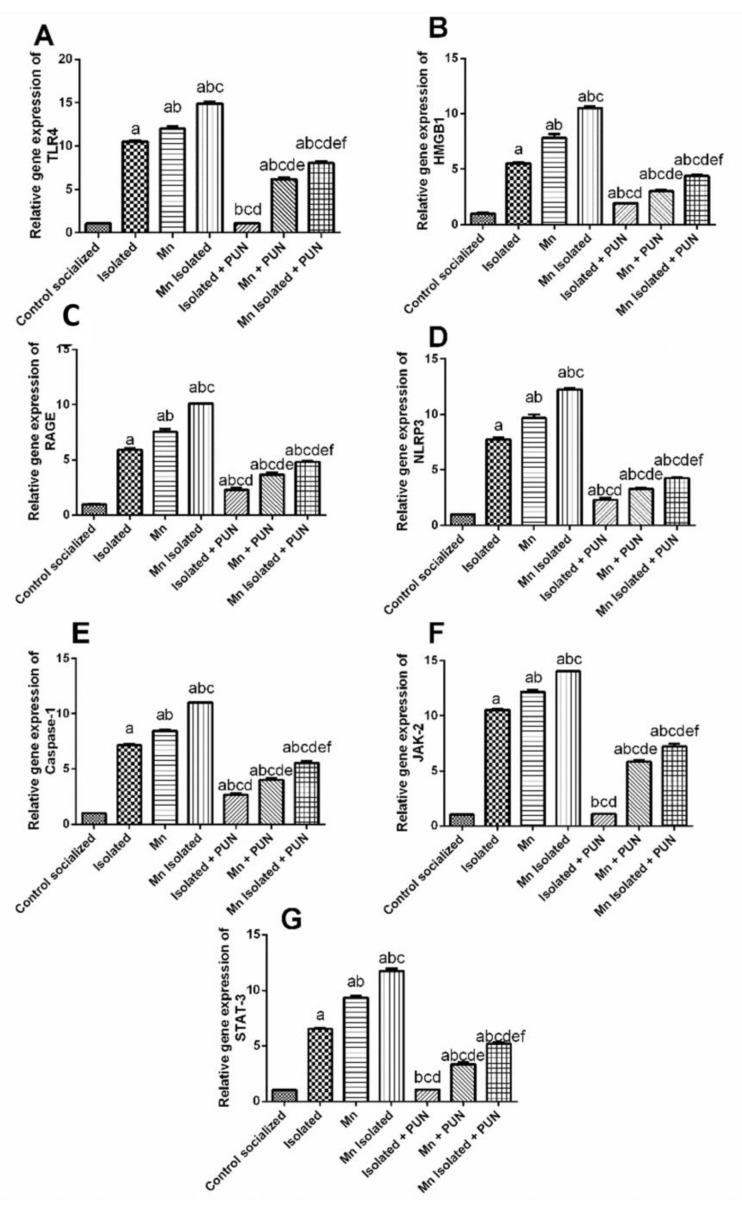
Impacts of PUN on HMGB1/RAGE/TLR4, NLRP3/Caspase-1 and JAK-2/STAT-3 pathways in SI and/or MnCl_2_-intoxicated rats. mRNA expression levels of (**A**) TLR4, (**B**) HMGB1, (**C**) RAGE, (**D**) NLRP3, (**E**) Caspase-1, (**F)** JAK-2 and (**G**) STAT-3. Control socialized: Normal control socialized rats, 4 per cage and fed rodent chow diet, Isolated: Rats were placed individually in laboratory cages, Mn: Rodents were treated daily with MnCl_2_ (10 mg/kg, i.p.) to induce PD, Mn Isolated: Socially isolated rats received MnCl_2_, Isolated + PUN: Socially isolated rats received PUN (30 mg/kg, p.o.) for 5 weeks, Mn + PUN: Rats received MnCl_2_ and PUN, Mn Isolated + PUN: Socially isolated rats received MnCl_2_ and PUN. Data are exhibited as mean (n = 6) ± S.E.M. Significant difference was calculated relative to ^(a)^ Normal control, ^(b)^ Isolated, ^(c)^ MnCl_2_, ^(d)^ Mn+ Isolated, ^(e)^ Isolated + PUN, and ^(f)^ Mn + PUN groups. The significance level at *p*-value of <0.05.

**Figure 7 pharmaceutics-15-02420-f007:**
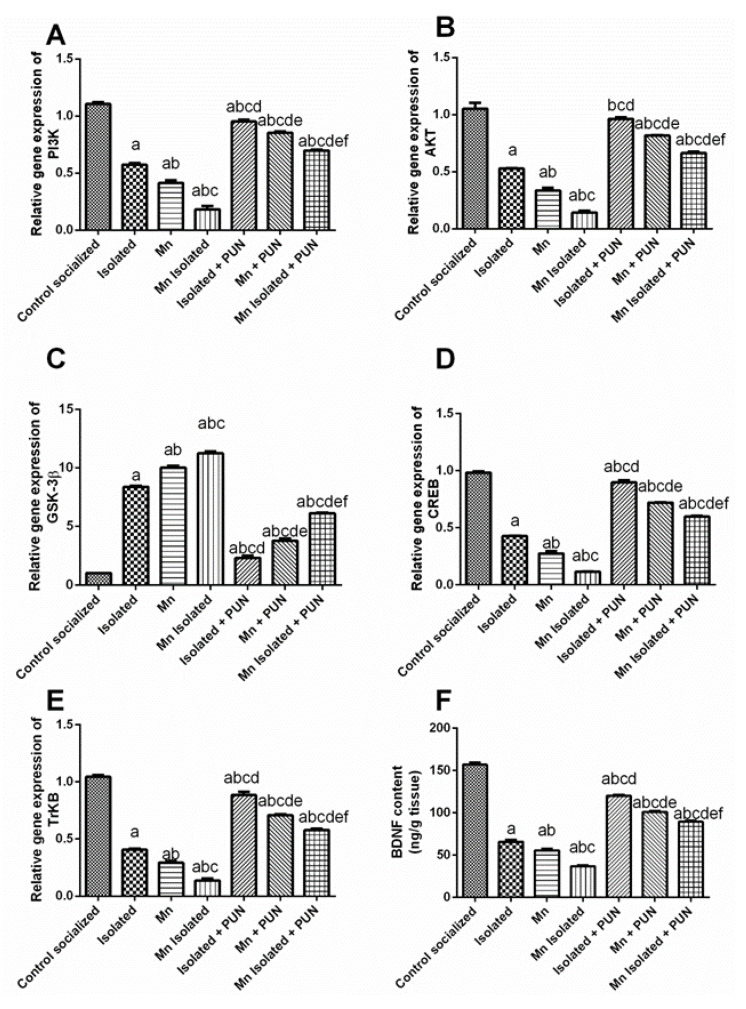
Impacts of PUN on PI3K/AKT/GSK-3β/CREB pathway in SI and/or MnCl_2_-intoxicated rats. (**A**) mRNA expression level of PI3K, (**B**) mRNA expression level of AKT, (**C**) mRNA expression level of GSK-3β, (**D**) mRNA expression level of CREB, (**E**) mRNA expression level of TrKB and (**F**) BDNF level. Control socialized: Normal control socialized rats, 4 per cage and fed rodent chow diet, Isolated: Rats were placed individually in laboratory cages, Mn: Rodents were treated daily with MnCl_2_ (10 mg/kg, i.p.) to induce PD, Mn Isolated: Socially isolated rats received MnCl_2_, Isolated + PUN: Socially isolated rats received PUN (30 mg/kg, p.o.) for 5 weeks, Mn + PUN: Rats received MnCl_2_ and PUN, Mn Isolated + PUN: Socially isolated rats received MnCl_2_ and PUN. Data are exhibited as mean (n = 6) ± S.E.M. Significant difference was calculated relative to ^(a)^ Normal control, ^(b)^ Isolated, ^(c)^ MnCl_2_, ^(d)^ Mn+ Isolated, ^(e)^ Isolated + PUN, and ^(f)^ Mn + PUN groups. The significance level at *p*-value of <0.05.

**Figure 8 pharmaceutics-15-02420-f008:**
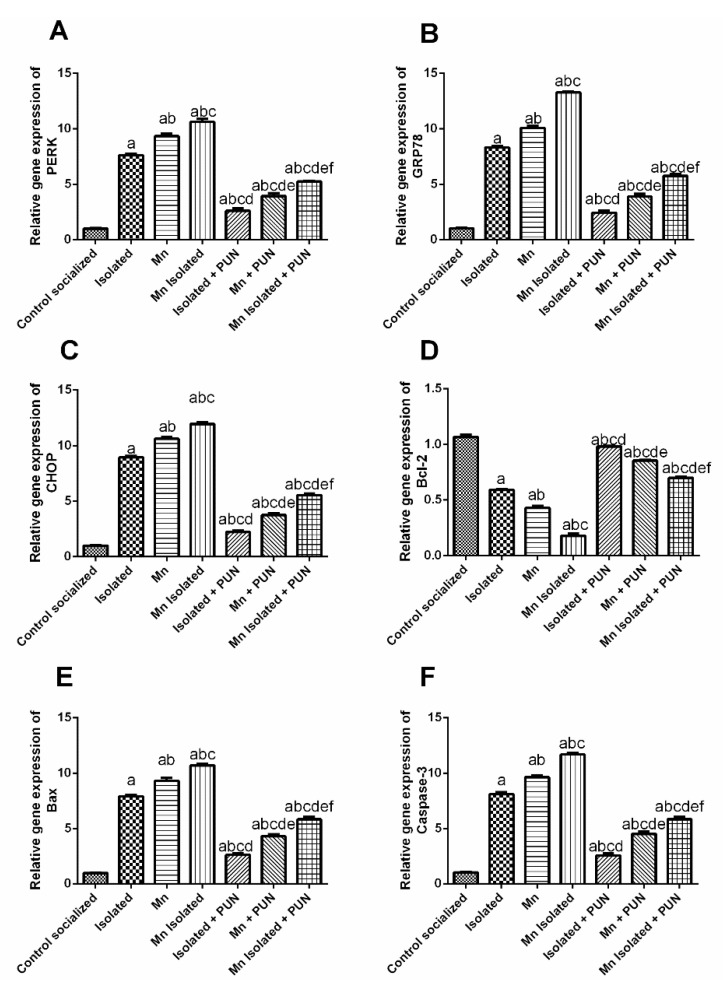
Impacts of PUN on endoplasmic reticulum stress biomarkers and apoptotic biomarkers in SI and/or MnCl_2_-intoxicated rats. mRNA expression levels of (**A**) PERK, (**B**) GRP78, (**C**) CHOP, (**D**) Bcl-2, (**E**) Bax, and (**F**) Caspase-3. Control socialized: Normal control socialized rats, 4 per cage and fed rodent chow diet, Isolated: Rats were placed individually in laboratory cages, Mn: Rodents were treated daily with MnCl_2_ (10 mg/kg, i.p.) to induce PD, Mn Isolated: Socially isolated rats received MnCl_2_, Isolated + PUN: Socially isolated rats received PUN (30 mg/kg, p.o.) for 5 weeks, Mn + PUN: Rats received MnCl_2_ and PUN, Mn Isolated + PUN: Socially isolated rats received MnCl_2_ and PUN. Data are exhibited as mean (n = 6) ± S.E.M. Significant difference was calculated relative to ^(a)^ Normal control, ^(b)^ Isolated, ^(c)^ MnCl_2_, ^(d)^ Mn+ Isolated, ^(e)^ Isolated + PUN, and ^(f)^ Mn + PUN groups. The significance level at *p*-value of <0.05.

**Figure 9 pharmaceutics-15-02420-f009:**
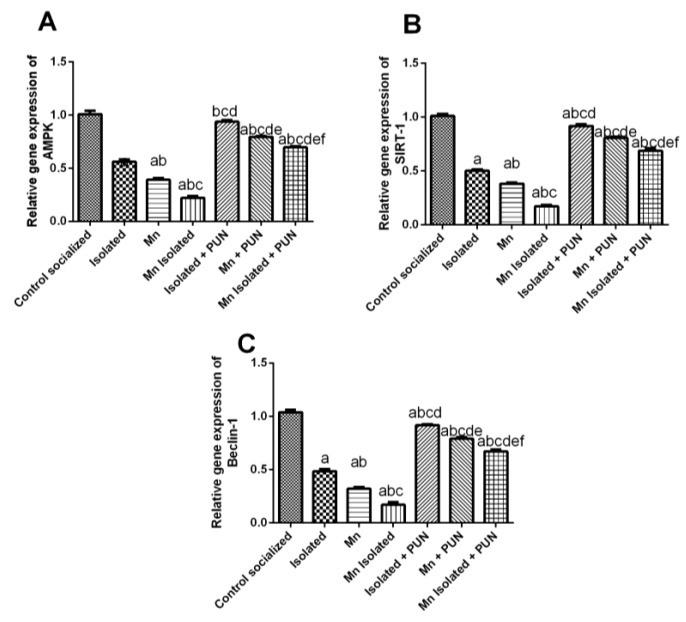
Impacts of PUN on AMPK/SIRT-1 pathway in SI and/or MnCl_2_-intoxicated rats. mRNA expression levels of (**A**) AMPK, (**B**) SIRT-1, and (**C**) Beclin-1. Control socialized: Normal control socialized rats, 4 per cage and fed rodent chow diet, Isolated: Rats were placed individually in laboratory cages, Mn: Rodents were treated daily with MnCl_2_ (10 mg/kg, i.p.) to induce PD, Mn Isolated: Socially isolated rats received MnCl_2_, Isolated + PUN: Socially isolated rats received PUN (30 mg/kg, p.o.) for 5 weeks, Mn + PUN: Rats received MnCl_2_ and PUN, Mn Isolated + PUN: Socially isolated rats received MnCl_2_ and PUN. Data are exhibited as mean (n = 6) ± S.E.M. Significant difference was calculated relative to ^(a)^ Normal control, ^(b)^ Isolated, ^(c)^ MnCl_2_, ^(d)^ Mn+ Isolated, ^(e)^ Isolated + PUN, and ^(f)^ Mn + PUN groups. The significance level at *p*-value of <0.05.

**Figure 10 pharmaceutics-15-02420-f010:**
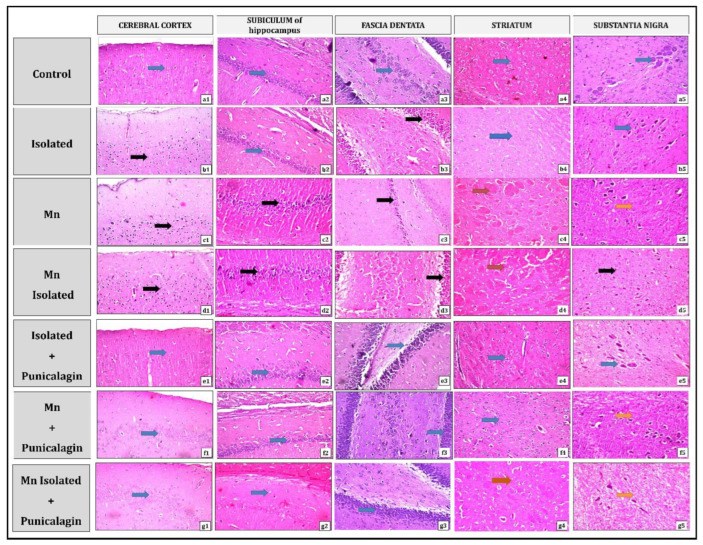
Photomicrographs of brain sections (cerebral cortex, subiculum, and fascia dentata in the hippocampus, striatum areas, and substantia nigra) stained by Hematoxylin and Eosin (magnification 40×). Representative photomicrographs of control group (**a1**–**5**), Isolated group (**b1**–**5**), MnCl_2_–intoxicated group (Mn) (**c1**–**5**), Mn Isolated group (**d1**–**5**), Isolated + PUN group (**e1**–**5**), Mn + PUN group (**f1**–**5**) and Mn Isolated + PUN group (**g1**–**5**). The blue arrow indicates no histopathological change, the black arrow shows nuclear pyknosis and degeneration, the orange arrow directs neurodegeneration, and the red arrow points to focal eosinophilic plagues. The control group did not exhibit histopathological modifications in any of the investigated brain areas (Inserts **a1**–**5**). However, the isolated group elicited nuclear pyknosis and degeneration in most neurons of each of the cerebral cortex (Insert **b1**) and fascia dentata (Insert **b3**), but no histopathological changes were shown in the subiculum, striatal areas, and substantia nigra (Inserts **b2**, **b4**, and **b5**). Moreover, Mn group showed marked nuclear pyknosis and degeneration in most neurons of each of the following, cerebral cortex (Insert **c1**), subiculum (Insert **c2**), and fascia dentata (Insert **c3**), while the striatal areas exhibited large-size focal eosinophilic plagues formation (Insert c4), and some neurons of substantia nigra showed degeneration (Insert **c5**). In Mn Isolated group, severe nuclear pyknosis and degeneration were remarked in most neurons of the cerebral cortex (Insert **d1**), together with all neurons of the subiculum (Insert **d2**) and most neurons in fascia dentata (Insert **d3**). Additionally, in this group, striatal areas exhibited marked creation of multiple large-size focal eosinophilic plagues with loss of neurons (Insert **d4**), whereas the substantia nigra showed severe nuclear pyknosis and degeneration in most of its neurons (Insert **d5**). Contrariwise, in the Isolated + PUN group, apparently there was no histopathological modification in the neurons of all investigated brain regions (Inserts **e1**–**5**). In addition, Mn + PUN group exhibited an apparent normal histological picture for neurons of each of the cerebral cortex (Insert **f1**), hippocampus (subiculum and fascia dentate) (Inserts **f2** and **f3**), and striatum (Insert **f4**), but showed slight degeneration in some neurons of substantia nigra (Insert **f5**). Furthermore, in Mn Isolated + PUN group, neurons of the cerebral cortex (Insert **g1**) and hippocampus (subiculum and fascia dentate) (Inserts **g2** and **g3**) revealed an apparent normal histological picture. Whereas the striatal area in this group showed a slight formation of focal fine plagues (Insert **g4**), and substantia nigra from the same group showed slight degeneration in some of its neurons (Insert **g5**).

**Table 1 pharmaceutics-15-02420-t001:** List of primer sequence sets utilized for RT-qPCR analysis in rat tissues.

Target Gene	Gene Forward and Backward Primer Sequence
Bax	F: 5′-ATGTTTTCTGACGGCAACTTC-3′ R: 5′-AGTCCAATGTCCAGCCCAT-3′
Bcl-2	F: 5′-CTACGAGTGGGATGCTGGAG-3′ R: 5′-TTCTTCACGATGGTGAGCG-3′
Caspase-3	F: 5′-GTGGAACTGACGATGATATGGC-3′ R: 5′-CGCAAAGTGACTGGATGAACC-3′
Caspase-1	F:5′-GAACAAAGAAGGTGGCGCAT-3′ R:5′-GAGGTCAACATCAGCTCCGA-3′
NLRP3	F:5′-TGCATGCCGTATCTGGTTGT-3′ R:5′-ACCTCTTGCGAGGGTCTTTG-3′
NF-Κβ	F:5′-CGCGGGGACTATGACTTGAA-3′ R:5′-AGTTCCGGTTTACTCGGCAG-3′
HO-1	F:5′-CACCAGCCACACAGCACTAC-3′ R: 5′-CACCCACCCCTCAAAAGACA-3′
Nrf2	F:5′-CTCTCTGGAGACGGCCATGACT-3′ R:5′-CTGGGCTGGGGACAGTGGTAGT-3′
TLR4	F: 5′-TCAGCTTTGGTCAGTTGGCT-3′ R: 5′-GTCCTTGACCCACTGCAAGA-3′
GSK-3β	F: 5′-ACACACCTGCCCTCTTCAAC-3′ R: 5′-GAAGCGGCGTTATTGGTCTG-3′
PERK	F: 5′-GCCGATGGGATAGTGATG-3′ R: 5′-GCAGCCTCTACAATGTCTTCT-3′
CHOP	F: 5′-TCTGCCTTTCGCCTTTGAG-3′ R: 5′-GCTTTGGGAGGTGCTTGTG-3′
GRP78	F:5′-GACATCAAGTTCTTGCCGTT-3′ R:5′-CTCATAACATTTAGGCCAGC-3′
PI3K	F: 5′-GCCCAGGCTTACTACAGAC-3′ R: 5′-AAGTAGGGAGGCATCTCG-3′
TrkB	F: 5′- CCTCCACGGATGTTGCTGA-3′ R: 5′-GGCTGTTGGTGATACCGAAGTA-3′
AMPK	F: 5′ -AAAGAACCCTAGCCTGAAGAGG-3′ R: 5′-ACCTTCCGAGATGAATGCTTTT-3′
SIRT 1	F: 5′- GGCACCGATCCTCGAACAAT-3′ R: 5′-CGCTTTGGTGGTTCTGAAAGG-3′
Beclin-1	F: 5′-AGCACGCCATGTATAGCAAAGA-3′ R:5′-GGAAGAGGGAAAGGACAGCAT-3′
AKT	F: 5′-AGGAGGAGGAGGAGATGGA-3′ R: 5′-GGTCGTGGGTCTGGAAAG-3′
CREB	F: 5′-CAGACAACCAGCAGAGTGGA-3′ R: 5′-CTGGACTGTCTGCCCATTG-3′
HMGB1	F: 5′-CACCCTGCATATTGTGGTAGG-3′ R: 5′-CGCTGGGACTAAGGTCAACA-3′
RAGE	F: 5′-GAGTCCGAGTCTACCAGATTCC-3′ R:5′-GGTCTCCTCCTTCACAACTGTC-3′
JAK-2	F: 5′-AGCTCCTCTCCTTGACGACT-3′ R:5′-GCACGCACTTCGGTAAGAAC-3′
STAT-3	F: 5′-CAAAGAAAACATGGCCGGCA-3′ R:5′-GGGGGCTTTGTGCTTAGGAT-3′
GAPDH	F:5′-AACTCCCATTCCTCCACCTT-3′ R:5-GAGGGCCTCTCTCTTGCTCT-3′

## Data Availability

Data will be available on request.

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
