# Peer review of "Punicalagin’s Protective Effects on Parkinson’s Progression in Socially Isolated and Socialized Rats: Insights into Multifaceted Pathway"

_pharmaceutics, 2023, doi:10.3390/pharmaceutics15102420_

Round 1

Reviewer 1 Report

The present manuscript “Protective effects of punicalagin against the progression of Parkinson’s disease in both socially isolated and socialized rats: Insights into the role of Nrf2/HO-1, HMGB1/RAGE/TLR4/NF-á´‹B, NLRP3/Caspase-1, JAK-2/STAT-3, PI3K/AKT/GSK-3β/CREB, AMPK/SIRT-1 and PERK/CHOP/Bcl-2 pathways” submitted by Salem and coworkers reports an interesting study in which the authors have evaluated the relationships between Social Isolation and Parkinson’s disease progression. In this work, the authors used a rat model of neurodegeneration based on the intoxication with MnCl2, which induces lasting motor and cognitive impairment in the animals. Using this model system, the authors have evaluated the effects of isolation and punicalagin administration on the progression of the disease. Punicalagin is a phenolic compound derived from plants with demonstrated anti-inflammatory and antioxidant properties.  While the data presented is interesting for the field, the manuscript is somehow preliminary and has some important issues that should be addressed.

Major comments and suggestions

  • Tittle. As currently written, the tittle of the manuscript is difficult to read. I would suggest the authors to provide a shorter version of the tittle.
  • Introduction section. This section is in general well-written. However, the current length is indeed excessive. I would suggest the authors to summarize this section and retain only the more relevant information.
  • Conclusions section. In a similar manner, the conclusion section comprises almost 5 pages. I would suggest the authors to provide a shorter version of discussion in the manuscript.
  • In this study, the authors used a MnCl2 intoxication model of the PD. They should discuss the reasons to use this model, instead other classical models of PD, for instance such as the 6-OHDA rat model.

Moderate editing of the English language is required.

Author Response

Dear Reviewer (1),

We sincerely appreciate the time and effort you dedicated to reviewing our manuscript titled "Protective Effects of Punicalagin against the Progression of Parkinson’s Disease in Both Socially Isolated and Socialized Rats: Insights into the Role of Nrf2/HO-1, HMGB1/RAGE/TLR4/NF-á´‹B, NLRP3/Caspase-1, JAK-2/STAT-3, PI3K/AKT/GSK-3β/CREB, AMPK/SIRT-1, and PERK/CHOP/Bcl-2 Pathways." Your valuable feedback and suggestions have significantly contributed to improving the quality and impact of our research.

Herein, we address your major comments and suggestions:

  1. **Title:** We appreciate your input regarding the complexity of our original title. We agreed with your suggestion and revised the title to reflect a more concise and focused representation of our study's core theme. We changed it into “Punicalagin's Protective Effects on Parkinson's Progression in Socially Isolated and Socialized Rats: Insights into Multifaceted Pathway”. We highlighted Changes with yellow.

  1. **Introduction Section:** We acknowledge your observation about the length of the introduction. In response, we have condensed the introduction, ensuring that it emphasizes the most relevant and critical information while providing an effective context for the study. We highlighted Changes with yellow.

  1. **Conclusions Section:** Your feedback on the length of the conclusions section was duly noted. To enhance the manuscript's overall flow, we revised the conclusion section to provide a succinct summary of our discussion points and key findings, delivering a more impactful closure.

  1. **Choice of Model:** We took your suggestion into consideration and elaborated on the rationale behind selecting the MnCl2 intoxication model over other classical models of Parkinson's disease, such as the 6-OHDA rat model. This explanation adds depth to our manuscript and clarifies the context of our model choice.

We are grateful for your guidance in shaping our work into a more refined and meaningful contribution to the field. Your thorough review has been invaluable in strengthening the manuscript's scientific content and presentation. Thank you for your commitment to advancing scientific knowledge and discourse.

Best regards,

Hoda Abdelaziz Salem

Department of Pharmacy practice, Faculty of Pharmacy, University of Tabuk, Tabuk 74191, Saudi Arabia

[email protected]

Reviewer 2 Report

The manuscript presents interesting results about the use of a natural compound as a therapy for PD. The experimental design and conclusions are well performed. However, some minor issues need to be fixed before publication. 

The title is too large and excessively descriptive. It should be shortened f.i to "Protective Effects of Punicalagin Against Parkinson's Disease Progression: Insights into Social Isolation and Involved Pathways in Rats". The rest of the information could be as keywords (as already).

Lacks type of paper (probably research)

Dopaminergic neurons do not deteriorate progressively, they die/disappear progressively.

G5-7 (groups treated with PUN corresponding to the four previous groups)--The abstract could specify that there is one group for each treatment condition, similar to the approach described in the Materials and Methods section, to avoid any confusion regarding grouping them.

Author Response

Dear Reviewer (2),

Thank you for dedicating your time to reviewing our manuscript titled "Protective Effects of Punicalagin against the Progression of Parkinson’s Disease in Both Socially Isolated and Socialized Rats: Insights into the Role of Nrf2/HO-1, HMGB1/RAGE/TLR4/NF-á´‹B, NLRP3/Caspase-1, JAK-2/STAT-3, PI3K/AKT/GSK-3β/CREB, AMPK/SIRT-1, and PERK/CHOP/Bcl-2 Pathways." We appreciate your positive evaluation of our experimental design and conclusions, and your insightful feedback for improvement.

We have considered your suggestions and made the necessary revisions:

  1. **Title: ** We have taken your recommendation to heart and revised the title to a more concise and clear version: " "Punicalagin's Protective Effects on Parkinson's Progression in Socially Isolated and Socialized Rats: Insights into Multifaceted Pathway" The remaining details have been included as keywords, as you suggested.
  2. **Type of Paper: ** Thank you for noting the absence of the paper type. We have now indicated that the manuscript is original research, ensuring the appropriate categorization.
  3. **Dopaminergic Neurons: ** We appreciate your correction. We have revised the relevant sections to accurately reflect that dopaminergic neurons die or disappear progressively, rather than deteriorating.
  4. **Grouping Clarification: ** We have revised the abstract to specify that there is one group for each treatment condition (G5-7), aligning with the approach detailed in the Materials and Methods section. This clarification should eliminate any confusion regarding the grouping structure.

Your insightful feedback has significantly contributed to enhancing the clarity and accuracy of our manuscript. We are grateful for your guidance and commitment to advancing the quality of scientific research. We are confident that these revisions will strengthen the manuscript and bring it closer to publication standards.

Best regards,

Hoda Abdelaziz Salem

Department of Pharmacy practice, Faculty of Pharmacy, University of Tabuk, Tabuk 74191, Saudi Arabia

[email protected]

Reviewer 3 Report

The author studied several signaling such as Nrf2/HO-1, HMGB1/RAGE/TLR4/NF-á´‹B, NLRP3/Caspase-1, JAK-2/STAT-3, PI3K/AKT/GSK-3β/CREB, AMPK/SIRT-1 in mRNA analysis.

In my opinion, why they did not show western blot (WB) data in the study. Is there any potential reason not examine all the protein in WB analysis? Need to explain it more preciously.

Overall, the study is unique and meaningful.

Author Response

Dear Reviewer (3),

We extend our gratitude for your thoughtful assessment of our manuscript. Your feedback on the signaling pathways studied, as well as the absence of Western blot (WB) data, is greatly appreciated and will undoubtedly aid in refining the manuscript.

Method: Western Blot* We would like to thank the reviewer for opening this critical point of research. We studied all pathways on both the mRNA level and the protein levels using qPCR (for mRNA) and different techniques for the proteins. The Activation of Nrf2/HO-1 pathway was confirmed not only by upregulation of their mRNA expression (by qPCR) but also by elevation detected in colorimetric measurement of SOD and TAC levels which is due to the reduction effect of the resultant proteins. Moreover, suppression of HMGB1/RAGE/TLR4/NFKB, NLRP3/Caspase-1 and JAK-2/STAT-3 pathways were verified by the reduction in levels of downstream inflammatory mediators in the active form of the proteins detected by ELISA measurements (IL-1β, prostaglandin E2 (PGE-2), inducible nitric oxide synthase (iNOS), IL-6, cyclooxygenase-2 (COX-2) and TNF-α). Additionally, stimulation of PI3K/AKT/GSK-3β/CREB/TrKB pathway was validated by elevation in BDNF protein level measured in the active form which was also measured using ELISA technique. All of these investigations focused on the active protein levels. Western blot assay is merely focusing on the protein level. The cellular environment is under the cellular stress. So, protein may be not in the active state. So, from our point of view, western blot may be not a good choice for measuring the protein activity. Hence, we preferred measuring the protein in the active form through (colorimetric and ELISA techniques) not by western blot. 

Thank you for recognizing the uniqueness and significance of our study. Your input has been invaluable in refining our work, and we remain committed to improving the manuscript based on your constructive comments.

Best regards,

Hoda Abdelaziz Salem

Department of Pharmacy practice, Faculty of Pharmacy, University of Tabuk, Tabuk 74191, Saudi Arabia

[email protected]

Round 2

Reviewer 1 Report

This is a revised version of the manuscript titled "Protective effects of punicalagin against the progression of Parkinson’s disease in both socially isolated and socialized rats: Insights into the role of Nrf2/HO-1, HMGB1/RAGE/TLR4/NF-á´‹B, NLRP3/Caspase-1, JAK-2/STAT-3, PI3K/AKT/GSK-3β/CREB, AMPK/SIRT-1 and PERK/CHOP/Bcl-2 pathways" submitted by Salem H. A. and coworkers. Having carefully reviewed the revised version of the manuscript, I am pleased to acknowledge that the authors have addressed all the minor and major drawbacks that were identified in the initial submission. Notably, they have made improvements in the title, introduction, and conclusions sections, which significantly enhanced the quality and clarity of the manuscript. Considering the comprehensive revisions made in response to my comments and other reviewers' feedback, my recommendation is publishing the manuscript in its present form.

Only minor editing of English language is required.